# Relationship between changing malaria burden and low birth weight in sub-Saharan Africa: A difference-in-differences study via a pair-of-pairs approach

Siyu Heng[1,2], Wendy P O'Meara[3], Ryan A Simmons[3,4], Dylan S Small[2]*

[1]Graduate Group in Applied Mathematics and Computational Science, School of Arts and Sciences, University of Pennsylvania, Philadelphia, United States; [2]Department of Statistics, The Wharton School, University of Pennsylvania, Philadelphia, United States; [3]Global Health Institute, School of Medicine, Duke University, Durham, United States; [4]Department of Biostatistics and Bioinformatics, School of Medicine, Duke University, Durham, United States

*For correspondence:
dsmall@wharton.upenn.edu

Competing interests: The authors declare that no competing interests exist.

## Abstract

**Background:** According to the World Health Organization (WHO), in 2018, an estimated 228 million malaria cases occurred worldwide with most cases occurring in sub-Saharan Africa. Scale-up of vector control tools coupled with increased access to diagnosis and effective treatment has resulted in a large decline in malaria prevalence in some areas, but other areas have seen little change. Although interventional studies demonstrate that preventing malaria during pregnancy can reduce the rate of low birth weight (i.e. child's birth weight <2500 g), it remains unknown whether natural changes in parasite transmission and malaria burden can improve birth outcomes.

**Methods:** We conducted an observational study of the effect of changing malaria burden on low birth weight using data from 18,112 births in 19 countries in sub-Saharan African countries during the years 2000–2015. Specifically, we conducted a difference-in-differences study via a pair-of-pairs matching approach using the fact that some sub-Saharan areas experienced sharp drops in malaria prevalence and some experienced little change.

**Results:** A malaria prevalence decline from a high rate (*Plasmodium falciparum* parasite rate in children aged 2-up-to-10 (i.e. $PfPR_{2-10}$) > 0.4) to a low rate ($PfPR_{2-10} < 0.2$) is estimated to reduce the rate of low birth weight by 1.48 percentage points (95% confidence interval: 3.70 percentage points reduction, 0.74 percentage points increase), which is a 17% reduction in the low birth weight rate compared to the average (8.6%) in our study population with observed birth weight records (1.48/8.6 ≈ 17%). When focusing on first pregnancies, a decline in malaria prevalence from high to low is estimated to have a greater impact on the low birth weight rate than for all births: 3.73 percentage points (95% confidence interval: 9.11 percentage points reduction, 1.64 percentage points increase).

**Conclusions:** Although the confidence intervals cannot rule out the possibility of no effect at the 95% confidence level, the concurrence between our primary analysis, secondary analyses, and sensitivity analyses, and the magnitude of the effect size, contribute to the weight of the evidence suggesting that declining malaria burden can potentially substantially reduce the low birth weight rate at the community level in sub-Saharan Africa, particularly among firstborns. The novel statistical methodology developed in this article–a pair-of-pairs approach to a difference-in-differences study–could be useful for many settings in which different units are observed at different times.

**Funding:** Ryan A. Simmons is supported by National Center for Advancing Translational Sciences (UL1TR002553). The funder had no role in study design, data collection and interpretation, or the decision to submit the work for publication.

## Introduction

In 2018, according to the *WHO, 2019* published by the WHO, an estimated 228 million malaria cases occurred worldwide, with an estimated 405,000 deaths from malaria globally (*WHO, 2019*). *Dellicour et al., 2010* estimated that around 85 million pregnancies occurred in 2007 in areas with stable *Plasmodium falciparum* (one of the most prevalent malaria parasites) transmission and therefore were at risk of malaria. Pregnant women are particularly susceptible to malaria, even if they have developed immunity from childhood infections, in part because parasitized cells in the placenta express unique variant surface antigens (*Rogerson et al., 2007*). Women who are infected during pregnancy may or may not experience symptoms, but the presence of parasites has grave consequences for both mother and unborn baby. Parasites exacerbate maternal anemia and they also sequester in the placenta, leading to intrauterine growth restriction, low birth weight (i.e. birth weight <2500 g), preterm delivery and even stillbirth and neonatal death. Preventing malaria during pregnancy with drugs or insecticide treated nets has a significant impact on pregnancy outcomes (*Eisele et al., 2012*; *Kayentao et al., 2013*; *Radeva-Petrova et al., 2014*).

Observational and interventional studies of malaria in pregnant women are complicated by the difficulty of enrolling women early in their pregnancy. However, in one study, early exposure to *Plasmodium falciparum* (before 120 days gestation), prior to initiating malaria prevention measures, was associated with a reduction in birth weight of more than 200 g and reduced average gestational age of nearly 1 week (*Schmiegelow et al., 2017*). For other representative studies on the negative influence of malaria infection during early pregnancy on birth outcomes, see *Menendez et al., 2000*, *Ross and Smith, 2006*, *Huynh et al., 2011*, *Valea et al., 2012*, *Walker et al., 2014*, and *Huynh et al., 2015*. These results suggest the impact of malaria infection on stillbirths, perinatal, and neonatal mortality may be substantial and needs more careful examination (*Fowkes et al., 2020*; *Gething et al., 2020*).

In the last few decades, malaria burden has declined in many parts of the world. Although the magnitude of the decline is difficult to estimate precisely, some reports suggest that the global cases of malaria declined by an estimated 41% between 2000 and 2015 (*WHO, 2016*) and the clinical cases of *Plasmodium falciparum* malaria declined by 40% in Africa between 2000 and 2015 (*Bhatt et al., 2015*). However, estimates of changing morbidity and mortality do not account for the effects of malaria in pregnancy. In the context of global reductions in malaria transmission, we expect fewer pregnancies are being exposed to infection and/or exposed less frequently. This should result in a significant reduction in preterm delivery, low birth weight and stillbirths. However, declining transmission will also lead to reductions in maternal immunity to malaria. Maternal immunity is important in mitigating the effects of malaria infection during pregnancy as is evidenced by the reduced impact of malaria exposure on the second, third and subsequent pregnancies. Thus, we anticipate a complex relationship between declining exposure and pregnancy outcomes that depends on both current transmission and historical transmission and community-level immunity (*Mayor et al., 2015*).

Understanding the potential causal effect of a reduction in malaria burden on the low birth weight rate is crucial as low birth weight is strongly associated with poor cognitive and physical development of children (*McCormick et al., 1992*; *Avchen et al., 2001*; *Guyatt and Snow, 2004*). Although we know from previous interventional studies that preventing malaria in pregnancy is associated with higher birth weight (*Eisele et al., 2012*; *Radeva-Petrova et al., 2014*), we do not know whether natural changes in malaria transmission intensity are similarly associated with improved birth outcomes. To address this question, we make use of the fact that while the overall prevalence of malaria has declined in sub-Saharan Africa, the decline has been uneven, with some malaria-endemic areas experiencing sharp drops and others experiencing little change. We use this heterogeneity to assess whether reductions in malaria prevalence reduce the proportion of infants born with low birth weight in sub-Saharan African countries. Our approach conducts a difference-in-differences study (*Card and Krueger, 2000*; *Angrist and Pischke, 2008*; *St. Clair and Cook, 2015*) by leveraging recent

**eLife digest** Malaria infects around 230 million people each year, mostly in sub-Saharan Africa, and causes more than 400,000 deaths. Pregnant women are particularly susceptible to malaria. The parasite that causes malaria can sap the mother's iron stores and may starve the baby of nutrients. Babies born to infected mothers often have low birth weights, which can have lasting effects on their health and brain development.

Previous studies suggest that preventing malaria in pregnant women using insecticide-treated bed nets or medications may improve birth outcomes. Successful efforts to prevent malaria have led to substantially fewer infections in sub-Saharan Africa. But success has been uneven with some communities continuing to have high rates of infection. These differences may allow scientists to better understand the community-level impact of falling rates of malaria on pregnancy outcomes in the real world.

Heng et al. estimated that reducing malaria transmission minimises the number of infants born with low birth weights in communities in sub-Saharan Africa. In an observational study, they used data on more than 18,000 births in 19 countries in this region between 2000 and 2015 to assess the effects of declining malaria rates on birth weights. They found that a decrease of malaria prevalence is estimated to reduce the rate of low birth weight by 1.48%, which is a 17% reduction in the number of observed newborns with low birth weight in the study population. First-born infants appeared to benefit the most.

This highlights that malaria interventions are beneficial for pregnant women and their newborns. Most analyses of the impact and cost-benefit of malaria control have ignored the potential advantages of malaria control on birth weight, and may thus undermine the benefits of malaria control. The approach used by Heng et al. may further be useful for other epidemiologists studying global health.

developments in matching, a nonparametric statistical analysis approach that can make studies more robust to bias that can arise from statistical model misspecification (*Rubin, 1973*; *Rubin, 1979*; *Hansen, 2004*; *Ho et al., 2007*).

## Materials and methods

### Overview

In this analysis, we combine two rich data sources: (1) rasters of annual malaria prevalence means (*Bhatt et al., 2015*) and (2) the Demographic and Health Surveys (DHS) (*ICF, 2019*), and we marry two powerful statistical analysis methods of adjusting for covariates – difference-in-differences (*Card and Krueger, 2000*; *Abadie, 2005*; *Athey and Imbens, 2006*; *Angrist and Pischke, 2008*; *Dimick and Ryan, 2014*; *St. Clair and Cook, 2015*) and matching (*Rubin, 1973*; *Rubin, 2006*; *Rosenbaum, 2002*; *Hansen, 2004*; *Stuart, 2010*; *Zubizarreta, 2012*; *Pimentel et al., 2015*). We match geographically proximal DHS clusters that were collected in different time periods (early vs. late) and then identify pairs of early/late clusters that have either maintained high malaria transmission intensity or experienced substantial declines in malaria transmission intensity. We then match pairs of clusters that differ in their malaria transmission intensity (maintained high vs. declined) but are similar in other key characteristics. Once these quadruples (pairs of pairs) have been formed, our analysis moves to the individual births within these clusters. We use multiple imputation to categorize missing children's birth weight records as either low birth weight or not, relying on the size of the child at birth reported subjectively by the mother and other demographic characteristics of the mother. Finally, we estimate the effect of declined malaria transmission intensity on the low birth weight rate by looking at the coefficient of the malaria prevalence indicator (low vs. high) contributing to the low birth weight rate in a mixed-effects linear probability model adjusted for covariates that are potential confounding variables, the group indicator (individual being within a cluster with declined vs. maintained high malaria transmission intensity), and the time indicator (late vs. early).

## Data resources

The data we use in this work comes from the following three sources:

1. Rasters of annual malaria prevalence: These image data, constructed by the Malaria Atlas Project (MAP) (*Hay and Snow, 2006*; *MAP, 2020*), estimate for sub-Saharan Africa the spatial distribution of the *Plasmodium falciparum* parasite rate (i.e. the proportion of the population that carries asexual blood-stage parasites) in children from 2 to 10 years old ($\mathrm{PfPR}_{2-10}$) for each year between 2000 and 2015 (*Bhatt et al., 2015*). $\mathrm{PfPR}_{2-10}$ has been widely used for measuring malaria transmission intensity (*Metselaar and Van Thiel, 1959*; *Smith et al., 2007*; *Bhatt et al., 2015*; *WHO, 2019*) and we use it in this work. The value in each pixel indicates the estimated annual $\mathrm{PfPR}_{2-10}$ (ranging from 0 to 1) with a resolution of 5 km by 5 km.

2. Demographic and Health Surveys (DHS): The DHS are nationally-representative household surveys mainly conducted in low- and middle- income countries that contain data with numerous health and sociodemographic indicators (*Corsi et al., 2012*; *ICF, 2019*). We used the Integrated Public Use Microdata Series' recoding of the DHS variables (IPUMS-DHS) which makes the DHS variables consistent across different years and surveys (*Boyle et al., 2019*).

3. Cluster Global Positioning System (GPS) data set: This data set contains the geographical information (longitude, latitude and the indicator of urban or rural) of each cluster in the DHS data. In order to maintain respondent confidentiality, the DHS program randomly displaces the GPS latitude/longitude positions for all surveys, while ensuring that the positional error of the clusters is at most 10 kilometers (at most 5 km for over 99% of clusters) and all the positions stay within the country and DHS survey region (*DHS, 2019*).

## Data selection procedure

In this article, we set the study period to be the years 2000–2015, and correspondingly, all the results and conclusions obtained in this article are limited to the years 2000–2015. We set the year 2000 as the starting point of the study period for two reasons. First, the year 2000 is the earliest year in which the estimated annual $\mathrm{PfPR}_{2-10}$ is published by *MAP, 2020*. Second, according to *Bhatt et al., 2015*, 'the year 2000 marked a turning point in multilateral commitment to malaria control in sub-Saharan Africa, catalysed by the Roll Back Malaria initiative and the wider development agenda around the United Nations Millennium Development Goals'. We set the year 2015 as the ending point based on two considerations. First, when we designed our study in the year 2017, the year 2015 was the latest year in which the estimated annual $\mathrm{PfPR}_{2-10}$ was available to us. We became aware after starting our outcome analysis that MAP has published some post-2015 estimated annual $\mathrm{PfPR}_{2-10}$ data since then, but, following *Rubin, 2007*'s advice to design observational studies before seeing and analyzing the outcome data, we felt it was best to stick with the design of our original study for this report and consider the additional data in a later report. Second, the year 2015 was set as a target year by a series of international goals on malaria control. For example, the United Nations Millennium Development Goals set a goal to 'halt by 2015 and begin to reverse the incidence of malaria' and 'the more ambitious target defined later by the World Health Organization (WHO) of reducing case incidence by 75% relative to 2000 levels.' (*WHO, 2008*; *Bhatt et al., 2015*). It is worth emphasizing that although we set the years 2000–2015 as the study period and did not investigate any post-2015 MAP data because of the above considerations, those published or upcoming post-2015 MAP data should be considered or leveraged for future related research or follow-up studies.

After selecting 2000–2015 as our study period, we take the middle point years 2007 and 2008 as the cut-off and define the years 2000–2007 as the 'early years' and the years 2008–2015 as the 'late years.' We include all the sub-Saharan countries that satisfy the following two criteria: (1) The rasters of estimated annual $\mathrm{PfPR}_{2-10}$ between 2000 and 2015 created by the Malaria Atlas Project include that country. (2) For that country, IPUMS-DHS contains at least one standard DHS between 2000 and 2007 ('early year') and at least one standard DHS between 2008 and 2015 ('late year'), and both surveys include the cluster GPS coordinates. If there is more than one early (late) years for which the above data are all available, we chose the earliest early year (latest late year). This choice was made to maximize the time interval for the decrease of malaria prevalence, if any, to have an effect on the birth weight of infants. For those countries that have at least one standard DHS with available cluster GPS data in the late year (2008–2015), but no available standard DHS or GPS data in the early year (2000–2007), we still include them if they have a standard DHS along with its GPS

data for the year 1999 (with a possible overlap into 1998). In this case, we assign MAP annual $\mathrm{PfPR}_{2-10}$ estimates from 2000 to the 1999 DHS data. This allows us to include two more countries, Cote d'Ivoire and Tanzania. The 19 sub-Saharan African countries that meet the above eligibility criteria are listed in *Table 1*.

From *Table 1*, we can see that among the 19 countries, only two countries (Congo Democratic Republic and Zambia) happen to take the margin year 2007 as the early year and no countries take the margin year 2008 as the late year. This implies that our study is relatively insensitive to our way of defining the early years (2000–2007) and the late years (2008–2015) as most of the selected early years and late years in *Table 1* do not fall near the margin years 2007 and 2008.

## Statistical analysis

### Motivation and overview of our approach: difference-in-differences via pair-of-pairs

Our approach to estimating the causal effect of reduced malaria burden on the low birth weight rate is to use a difference-in-differences approach (*Card and Krueger, 2000*; *Abadie, 2005*; *Athey and Imbens, 2006*; *Angrist and Pischke, 2008*; *Dimick and Ryan, 2014*; *St. Clair and Cook, 2015*) combined with matching (*Rubin, 1973*; *Rosenbaum, 2002*; *Hansen, 2004*; *Stuart, 2010*; *Zubizarreta, 2012*; *Pimentel et al., 2015*). In a difference-in-differences approach, units are measured in both an early (before treatment) and late (after treatment) period. Ideally, we would like to observe how the low birth weight rate changes with respect to malaria prevalence within each DHS cluster, so that the DHS clusters themselves could be the units in a difference-in-differences approach. However, this is not feasible because within each country over time the DHS samples different locations (clusters) as the representative data of that country. We use optimal matching (*Rosenbaum, 1989*; *Hansen and Klopfer, 2006*) to pair two DHS clusters, one in the early year and one in the late year as closely as possible, mimicking a single DHS cluster measured twice in two different time periods.

**Table 1.** The 19 selected sub-Saharan African countries along with their chosen early/late years of malaria prevalence (i.e. estimated parasite rate $\mathrm{PfPR}_{2-10}$) and IPUMS-DHS early/late years.
Note that some DHS span over two successive years.

| Country | Malaria prevalence | | IPUMS-DHS | |
|---|---|---|---|---|
| | Early year | Late year | Early year | Late year |
| Benin | 2001 | 2012 | 2001 | 2011–12 |
| Burkina Faso | 2003 | 2010 | 2003 | 2010 |
| Cameron | 2004 | 2011 | 2004 | 2011 |
| Congo Democratic Republic | 2007 | 2013 | 2007 | 2013–14 |
| Cote d'Ivoire | 2000 | 2012 | 1998–99 | 2011–12 |
| Ethiopia | 2000 | 2010 | 2000 | 2010–11 |
| Ghana | 2003 | 2014 | 2003 | 2014 |
| Guinea | 2005 | 2012 | 2005 | 2012 |
| Kenya | 2003 | 2014 | 2003 | 2014 |
| Malawi | 2000 | 2010 | 2000 | 2010 |
| Mali | 2001 | 2012 | 2001 | 2012–13 |
| Namibia | 2000 | 2013 | 2000 | 2013 |
| Nigeria | 2003 | 2013 | 2003 | 2013 |
| Rwanda | 2005 | 2014 | 2005 | 2014–15 |
| Senegal | 2005 | 2010 | 2005 | 2010–11 |
| Tanzania | 2000 | 2015 | 1999 | 2015–16 |
| Uganda | 2000 | 2011 | 2000–01 | 2011 |
| Zambia | 2007 | 2013 | 2007 | 2013–14 |
| Zimbabwe | 2005 | 2015 | 2005–06 | 2015 |

After this first-step matching, we define the treated units as the high-low pairs of clusters, meaning that the early year cluster has high malaria prevalence (i.e. $\text{PfPR}_{2-10} > 0.4$) while the late year cluster has low malaria prevalence (i.e. $\text{PfPR}_{2-10} < 0.2$), and define the control units as the high-high pairs of clusters, meaning that both the early year and late year clusters have high malaria prevalence (i.e. $\text{PfPR}_{2-10} > 0.4$) and the absolute difference between their two values of $\text{PfPR}_{2-10}$ (one for the early year and one for the late year) is less than 0.1. The difference-in-differences approach (*Card and Krueger, 2000*; *Angrist and Pischke, 2008*; *Dimick and Ryan, 2014*; *St. Clair and Cook, 2015*) compares the changes in the low birth weight rate over time for treated units (i.e. high-low pairs of clusters) compared to control units (i.e. high-high pairs of clusters) adjusted for observed covariates. The difference-in-differences approach removes bias from three potential sources (*Volpp et al., 2007*):

- A difference between treated units and control units that is stable over time cannot be mistaken for an effect of reduced malaria burden because each treated or control unit is compared with itself before and after the time at which reduced malaria burden takes place in the treated units.
- Changes over time in sub-Saharan Africa that affect all treated or control units similarly cannot be mistaken for an effect of reduced malaria burden because changes in low birth weight over time are compared between the treated units and control units.
- Changes in the characteristics (i.e. observed covariates) of the populations (e.g. age of mother at birth) in treated or control units over time cannot be mistaken for an effect of reduced malaria burden as long as those characteristics are measured and adjusted for.

The traditional difference-in-differences approach requires a parallel trend assumption, which states that the path of the outcome (e.g. the low birth weight rate) for the treated unit is parallel to that for the control unit (*Card and Krueger, 2000*; *Angrist and Pischke, 2008*; *Dimick and Ryan, 2014*; *St. Clair and Cook, 2015*). One way the parallel trend assumption can be violated is if there are events in the late period whose effect on the outcome differs depending on the level of observed covariates and those observed covariates are unbalanced between the treated and control units across time (*Shadish et al., 2002*). For example, suppose that there are advances in prenatal care in the late year that tend to be available more in urban areas, then the parallel trends assumption could be violated if there are more treated units (i.e. high-low pairs of clusters) in urban areas than control units (i.e. high-high pairs of clusters). To make the parallel trend assumption more likely to hold, instead of conducting a difference-in-differences study simply among all the treated and control units, we use a second-step matching to pair treated units with control units on the observed covariates trajectories (from the early year to the late year) to make the treated units and control units similar in the observed covariates trajectories as they would be under randomization (*Rosenbaum, 2002*; *Stuart, 2010*), and discard those treated or control units that cannot be paired with similar observed covariates trajectories. For example, by matching on the urban/rural indicator trajectories between the treated and control units, we adjust for the potential source of bias resulting from the possibility that there may be advances in prenatal care in the late year that are available more in urban areas.

Another perspective on how our second-step matching helps to improve a difference-in-differences study is through the survey location sampling variability (*Fakhouri et al., 2020*). Recall that when constructing representative samples, the DHS are sampled at different locations (i.e. clusters) across time (*ICF, 2019*; *Boyle et al., 2019*). Therefore, if we simply implemented a difference-in-differences approach over all the high-low and high-high pairs of survey clusters and did not use matching to adjust for observed covariates, this survey location sampling variability may generate imbalances (i.e. different trajectories) of observed covariates across the treated and control groups, and therefore may bias the difference-in-differences estimator (*Heckman et al., 1997*). Imbalances of observed covariates caused by the survey location sampling variability may occur in the following three cases: (1) The survey location sampling variability is affecting the treated and control groups in the opposite direction. Specifically, there is some observed covariate for which the difference between the high-low pairs of sampled clusters tends to be larger (or smaller) than the country's overall difference between the high malaria prevalence regions in the early years and the low malaria prevalence regions in the late years and conversely, the difference in that observed covariate between the high-high pairs of sampled clusters tends to be smaller (or larger) than the country's overall difference

between the high malaria prevalence regions in the early years and the high malaria prevalence regions in the late years. (2) The survey location sampling variability is affecting the treated and control groups in the same direction but to different extents. (3) The survey location sampling variability only happened in the treated or control group. Specifically, there is some observed covariate for which the difference between the high-low (or high-high) pairs of sampled clusters tends to differ from the country's overall difference between the high malaria prevalence regions in the early years and the low (or high) malaria prevalence regions in the late years, but this is not the case for the high-high (or high-low) pairs of sampled clusters. Using matching as a nonparametric data preprocessing step in a difference-in-differences study can remove this type of bias because the observed covariates trajectories are forced to be common among the matched treated and control groups (*St. Clair and Cook, 2015*; *Basu and Small, 2020*).

An additional important aspect of our approach is that we use multiple imputation to address missingness in the birth weight records. The fraction of missingness in birth weight in the IPUMS-DHS data set is non-negligible and previous studies have noted that failing to carefully and appropriately address the missing data issue with the birth weight records can significantly bias the estimates of the low birth weight rate derived from surveys in developing countries (*Boerma et al., 1996*; *Robles and Goldman, 1999*). We address the missing data issue by using multiple imputation with carefully selected covariates. Multiple imputation constructs several plausible imputed data sets and appropriately combines results obtained from each of them to obtain valid inferences under an assumption that the data is missing at random conditional on measured covariates (*Rubin, 1987*). Our workflow is summarized in *Figure 1*, in which we indicate both the data granularity (country-level, cluster-level, and individual-level) and the corresponding steps of our statistical methodology (including the data selection procedure described in the previous section and the Steps 1–4 of the statistical analysis listed below).

## Step 1: Proximity prioritized in the matching of high-high and high-low clusters

The DHS collects data from different clusters within the same country in different survey years. To construct pairs of early year and late year clusters which are geographically close such that each pair of clusters can mimic a single cluster measured twice in two different time periods to serve as the unit of a difference-in-differences study, we use optimal matching (*Rosenbaum, 1989*; *Hansen and Klopfer, 2006*) to pair clusters within the same country, one from the early year and one from the late year, based on the geographic proximity of their locations. Specifically, we minimize the total rank-based Mahalanobis distance based on the latitude and longitude of the cluster with a propensity score caliper to pair clusters so that the total distance between the paired early year cluster and late year cluster is as small as possible (*Rosenbaum, 1989*; *Hansen and Klopfer, 2006*). The number of clusters to pair for each country is set to be the minimum of the number of clusters in the early year and the number of clusters in the late year of that country.

## Step 2: Matching on sociodemographic similarity is emphasized in second matching

We first divide malaria prevalence into three levels with respect to the estimated *Plasmodium falciparum* parasite rates $\text{PfPR}_{2-10}$ (ranging from 0 to 1): high ($\text{PfPR}_{2-10} > 0.4$), medium ($\text{PfPR}_{2-10}$ lies in [0.2, 0.4]), and low ($\text{PfPR}_{2-10} < 0.2$). For clusters in the year 1999, we use the $\text{PfPR}_{2-10}$ in the nearest year in which it is available, that is, the year 2000. We select the pairs of the early year and late year clusters as formed in Step 1 described above that belong to either one of the following two categories: (1) High-high pairs: both of the estimated parasite rates of the early year and late year clusters within that pair are high (>0.4), and the absolute difference between the two rates is less than 0.1. (2) High-low pairs: the estimated parasite rate of the early year cluster within that pair is high (>0.4), while the estimated parasite rate of the late year cluster within that pair is low (<0.2). A total of 950 out of 6812 pairs of clusters met one of these two criteria with 540 being high-high pairs and 410 high-low pairs. We removed one high-low pair in which the late year cluster had an estimated parasite rate value (i.e. $\text{PfPR}_{2-10}$) of zero for every year between 2000 and 2015; this cluster was in a high altitude area with temperature unsuitable for malaria transmission and thus was not comparable in malaria transmission intensity to its paired early year cluster with high malaria transmission

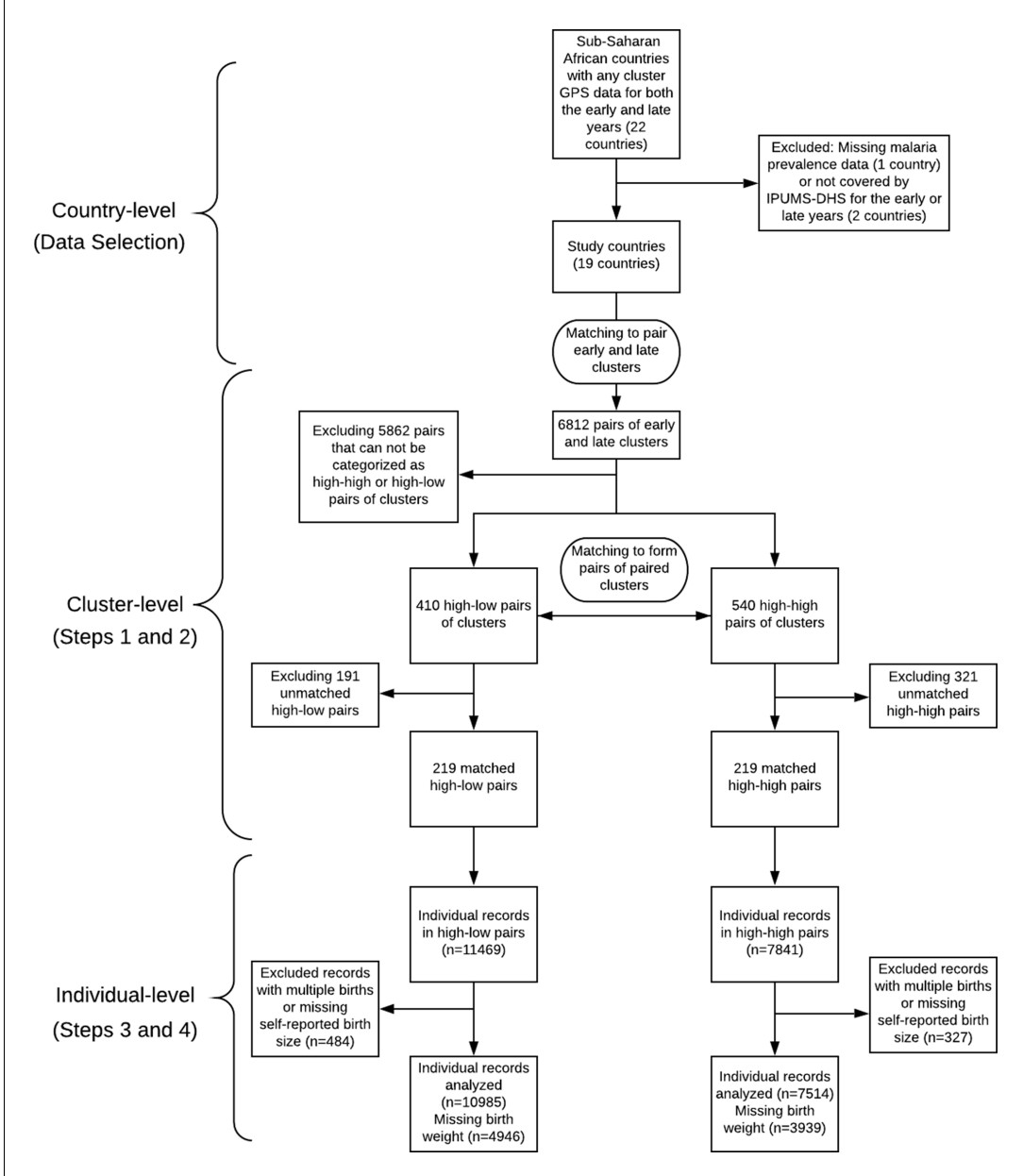

**Figure 1.** Work flow diagram of the study.

intensity. Since we would like to study the effect of reduced malaria burden on the low birth weight rate of infants, we consider high-low pairs of clusters as treated units and high-high pairs of clusters as control units, and conduct a matched study by matching each high-low pair with a high-high pair that is similar with respect to covariates that might be correlated with either the treatment (changes in malaria prevalence) or the outcome (low birth weight). We allow matches across different countries. The covariates we match on are cluster averages of the following individual-level covariates, where we code the individual-level covariates as quantitative variables with higher values suggesting higher sociodemographic status:

- Household electricity: 0 – dwelling has no electricity; 1 – otherwise.
- Household main material of floor: 1 – natural or earth-based; 2 – rudimentary; 3 – finished.
- Household toilet facility: 0 – no facility; 1 – with toilet.
- Urban or rural: 0 – rural; 1 – urban.
- Mother's education level: 0 – no education; 1 – primary; 2 – secondary or higher.

- Indicator of whether the woman is currently using a modern method of contraception: 0 – no; 1 – yes.

The above six sociodemographic covariates were chosen by looking over the variables in the Demographic and Health Surveys (DHS) and choosing those which we thought met the following criteria: (1) The above six covariates are potentially strongly correlated with both the risk of malaria (*Baragatti et al., 2009*; *Krefis et al., 2010*; *Ayele et al., 2013*; *Roberts and Matthews, 2016*; *Sulyok et al., 2017*) and birth outcomes (*Sahn and Stifel, 2003*; *Gemperli et al., 2004*; *Chen et al., 2009*; *Grace et al., 2015*; *Padhi et al., 2015*), and therefore may be important confounding variables that need to be adjusted for via statistical matching (*Rosenbaum and Silber, 2009*; *Rosenbaum, 2010*; *Stuart, 2010*). (2) The records of the above six covariates are mostly available for all the countries and the survey years in our study samples. Specifically, for the above six covariates, the percentages of missing data (missingness can arise either because the question was not asked or the individual was asked the question but did not respond) among the total individual records from IPUMS-DHS among the 6812 pairs of clusters remaining after Step 1 are all less than 0.3%.

For each cluster, we define the corresponding six cluster-level covariates by taking the average value for each of the six covariates among the individual records from IPUMS-DHS which are in that cluster, leaving out all missing data. This method of building up cluster-level data from individual-level records from DHS has been commonly used (*Kennedy et al., 2011*; *Larsen et al., 2017*). We form quadruples (pairs of pairs) by pairing one high-low pair of clusters (a 'treated' unit) with one high-high pair of clusters (a 'control' unit), such that all the six cluster-level observed covariates are balanced between both the early and late year clusters for the paired high-low and high-high pairs. We use optimal cardinality matching to form these quadruples (*Zubizarreta et al., 2014*; *Visconti and Zubizarreta, 2018*). Optimal cardinality matching is a flexible matching algorithm which forms the largest number of pairs of treated and control units with the constraint that the absolute standardized differences (absolute value of difference in means in standard deviation units; see *Rosenbaum, 2010*) are less than a threshold; we use a threshold of 0.1, which is commonly used to classify a match as adequate (*Neuman et al., 2014*; *Silber et al., 2016*). After implementing the optimal cardinality matching, 219 matched quadruples (pairs of high-low and high-high pairs of clusters) remain. See *Figure 2* for illustration of the process of forming matched quadruples (pairs of pairs).

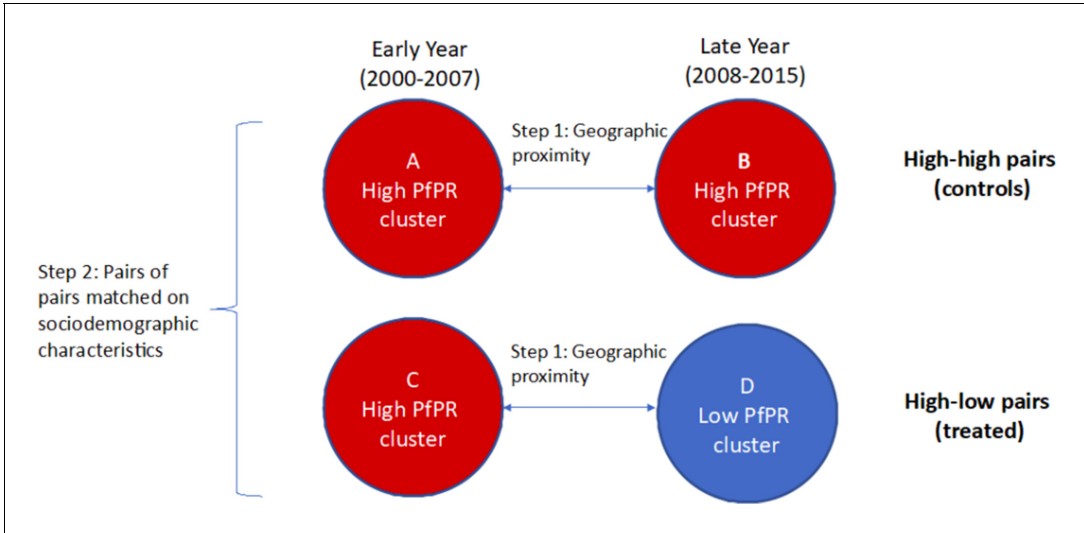

**Figure 2.** Formed quadruples (pairs of pairs) of matched high-low and high-high pairs of clusters. In Step 1, pairs of clusters from the early and late time periods are matched on geographic proximity and categorized as 'high-high' (comparison, or control) or 'high-low' (treated). In Step 2, pairs of high-high clusters are matched with pairs of high-low clusters based on cluster-level sociodemographic characteristics. The difference-in-differences estimate of the coefficient of changing malaria burden on the low birth weight rate is based on comparing (D–C) to (B–A).

## Step 3: Low birth weight indicator with multiple imputation to address missingness

We then conduct statistical analysis at the individual child level. Among all the 19,310 children's records from the quadruples formed above, we exclude multiple births (i.e. twins, triplets etc), leaving 18,499 records. The outcome variable is the indicator of low birth weight, which is defined as child's birth weight less than 2500 g. However, 48% of the birth weight records of children among these 18,499 records are missing. To handle this, we perform multiple imputation, under the assumption of missing at random (*Heitjan and Basu, 1996*), with 500 replications. An important predictor that is available for imputing the missing low birth weight indicator is the mother's subjective reported size of the child. The mother's reported size of the child is relatively complete in the IPUMS-DHS data set and has been shown to be a powerful tool to handle the missing data problem with birth weight (*Blanc and Wardlaw, 2005*). We exclude the small number of records with missing mother's subjective reported size of the child, leaving 18,112 records, 47% of which (8509 records) have missing low birth weight indicator. Among the 9603 records with observed birth weight, 825 (8.6%) had low birth weight. We first use the bayesglm function (part of the arm package in R) to fit a Bayesian logistic regression for the outcome of the low birth weight indicator among those children for whom low birth weight is not missing. To make it more plausible that the missing at random assumption holds, the following covariates are included as predictors in this regression because they might affect both missingness and the low birth weight rate:

- The size of the child at birth reported subjectively by the mother: 1 – very small or smaller than average; 2 – average; 3 – larger than average or very large.
- Mother's age in years.
- Child's birth order number: 1 – the first child born to a mother; 2 – the second, third or fourth child born to a mother; 3 – otherwise.
- Household wealth index: 1 – poorest; 2 – poorer; 3 – middle; 4 – richer; 5 – richest.
- Urban or rural: 0 – rural; 1 – urban.
- Mother's education level: 0 – no education; 1 – primary; 2 – secondary or higher.
- Child's sex: 0 – female; 1 – male.
- Mother's current marital or union status: 0 – never married or formerly in union; 1 – married or living together.
- Indicator of whether the child's mother received any antenatal care while the child was in utero: 0 – no or missing; 1 – yes.

We also include quadratic terms for mother's age in years and child's birth order in the regression since according to *Selvin and Janerich, 1971*, the influences of mother's age and child's birth order on the birth weight do not follow a linear pattern. Note that among the remaining 18,112 records, there are no missing data for all of the above covariates. The prior distributions for the regression coefficients follow the default priors of the bayesglm function, that is, independent Cauchy distributions with center 0 and scale set to 10 for the regression intercept term, 2.5 for binary predictors, and $2.5/(2 \times \text{sd})$ for other numerical predictors, where sd is the standard deviation of the predictor in the data used for fitting the regression (i.e. the 9603 records with observed birth weight). This default weakly informative prior has been shown to outperform Gaussian and Laplacian priors in a wide variety of settings (*Gelman et al., 2008*). After fitting this Bayesian logistic regression model, we get the posterior distribution of the regression coefficient associated with each predictor; see *Table 2*. From *Table 2*, we can see that in the imputation model, mother's age, child's birth order, mother's education level, and the mother's reported birth size are significant predictors, which agrees with the previous literature (e.g. *Fraser et al., 1995*; *Strobino et al., 1995*; *Richards et al., 2001*; *Valero De Bernabé et al., 2004*).

We then conduct the following procedure in each run of multiple imputation. For each individual with missing birth weight, we first draw from the posterior distribution of the regression coefficients in *Table 2*, we then use these regression coefficients and the individual's covariates (as predictors) to find the probability of the individual having low birth weight and then we use this probability to randomly draw a low birth weight indicator for the individual. We conduct this procedure 500 times, getting 500 independent data sets with imputed low birth weight indicators.

**Table 2.** Summary of the Bayesian logistic regression model fitted over records with observed birth weight which is used to predict missing low birth weight indicators.

| Predictor | Posterior mean | Posterior std | z-score | p-value |
|---|---|---|---|---|
| (Intercept) | 1.916 | 0.628 | 3.051 | 0.002** |
| Mother's age (linear term) | −0.207 | 0.045 | −4.562 | <0.001*** |
| Mother's age (quadratic term) | 0.003 | 0.001 | 3.987 | <0.001*** |
| Wealth index | 0.060 | 0.037 | 1.591 | 0.112 |
| Child's birth order (linear term) | −0.989 | 0.338 | −2.925 | 0.003** |
| Child's birth order (quadratic term) | 0.211 | 0.086 | 2.447 | 0.014* |
| 0 - rural; 1 - urban | 0.126 | 0.103 | 1.214 | 0.225 |
| Mother's education level | −0.226 | 0.062 | −3.633 | <0.001*** |
| Child is boy | −0.068 | 0.083 | −0.815 | 0.415 |
| Mother is married or living together | −0.173 | 0.117 | −1.482 | 0.138 |
| Indicator of antenatal care | −0.046 | 0.093 | −0.493 | 0.622 |
| Indicator of low birth size | 2.410 | 0.090 | 26.776 | <0.001*** |
| Indicator of large birth size | −1.387 | 0.129 | −10.786 | <0.001*** |

## Step 4: Estimation of causal effect of reduced malaria burden on the low birth weight rate

For each of the 500 imputed data sets, we then fit a mixed-effects linear probability model (using the lmer function in the R package lme4) where there is a random effect (random intercept) for each cluster to account for the potential correlations between the outcomes among the individual records within the same cluster (*Gałecki and Burzykowski, 2013*). We include in the model the covariates which might be related to both whether an individual is in a high-low vs. high-high pair of clusters and the low birth weight rate. Specifically we include the predictors from the Bayesian logistic regression for multiple imputation as covariate regressors in the mixed-effects linear probability model (listed in *Table 2*), except for the mother's reported birth size. We do not include reported birth size because it is not a pretreatment variable and is a proxy for the outcome (*Rosenbaum, 1984*). In addition to the above covariates, we include in the model the following three indicators: (1) Low malaria prevalence indicator: indicates whether the individual is from a cluster with a low malaria prevalence ($\mathrm{PfPR}_{2-10} < 0.2$). (2) Time indicator: 0 – if the individual is from a early year cluster; 1 – if the individual is from a late year cluster. (3) Group indicator: 0 – if the individual is from a cluster in a high-high pair of clusters; 1 – if the individual is from a cluster in a high-low pair of clusters. Through adjusting for the time varying covariates via matching and including the above three indicators in the regression, our study uses a difference-in-differences approach for a matched observational study (*Wing et al., 2018*). Note that even though we do not explicitly incorporate matching into the final model (i.e. the mixed-effects linear probability model [1]), matching still reduces the bias due to potential statistical model misspecification in our analysis by being a non-parametric data preprocessing step which makes the distributions of the observed covariates of the selected treated and control units identical or similar, lessening the dependence of the results on the model used to adjust for the observed covariates (*Hansen, 2004*; *Ho et al., 2007*). Let $\mathbb{1}(A)$ be the indicator function of event $A$ such that $\mathbb{1}(A) = 1$ if $A$ is true and $\mathbb{1}(A) = 0$ otherwise. To conclude, we consider the following mixed-effects linear probability model for the individual $j$ in cluster $i$:

$$\mathbb{P}(Y_{ij} = 1 \mid i, \mathbf{X}_{ij}) = k_0 + k_1 \cdot \mathbb{1}(i \text{ is a low malaria prevalence cluster}) + k_2 \cdot \mathbb{1}(i \text{ is a late year cluster})$$
$$+ k_3 \cdot \mathbb{1}(i \text{ is from a high} - \text{low pair of clusters}) + \beta^T \mathbf{X}_{ij}, \qquad (1)$$
$$\text{with two error terms } \alpha_i \sim \mathcal{N}(0, \sigma_0) \text{ and } \epsilon_{ij} \sim \mathcal{N}(0, \sigma_1).$$

In Model (1), $Y_{ij}$ is the observed outcome (i.e. the low birth weight indicator) and $\mathbf{X}_{ij}$ the covariate regressors (including the quadratic terms of mother's age and child's birth order) of the individual $j$ in cluster $i$, and $\alpha_i$ is the random effect for cluster $i$. See *Table 3* for an interpretation of the coefficients of the three indicators and the intercept term (i.e. the $k_0, k_1, k_2, k_3$) within each matched

**Table 3.** An interpretation of the coefficients of the intercept term and the three indicators defined in model (1) (i.e. the $k_0, k_1, k_2, k_3$) within each matched quadruple.

The coefficient of the low malaria prevalence indicator (i.e. the $k_1$) incorporates the information of the magnitude of the effect of changing malaria burden (from high to low) on the low birth weight rate.

| Cluster | Prevalence | Time | Pair | Coefficients | Within-pair Contrast | Between-pair Contrast |
|---|---|---|---|---|---|---|
| 1 | High | Early | High-low | $k_0 + k_3$ | $k_1 + k_2$ | $k_1$ |
| 2 | Low | Late | High-low | $k_0 + k_1 + k_2 + k_3$ | | |
| 3 | High | Early | High-high | $k_0$ | $k_2$ | |
| 4 | High | Late | High-high | $k_0 + k_2$ | | |

quadruple. The estimated causal effect of reduced malaria burden (low vs. high malaria prevalence) on the low birth weight rate is the mean value of the 500 estimated coefficients on the low malaria prevalence indicator obtained (i.e. the $k_1$) from 500 runs of the mixed-effects linear regression described above. See Appendix 2 for more details on the statistical inference procedure with multiple imputation, which are also referred to as Rubin's rules (*Carpenter and Kenward, 2012*).

It is worth clarifying that although we take a Bayesian approach when imputing (i.e. predicting) the missing low birth weight indicators in Step 3 (i.e. imputation model) and then take a frequentist approach when conducting the 500 separate outcome analyses with the 500 imputed data sets in Step 4 (i.e. substantive model), these two different statistical perspectives (i.e. Bayesian and frequentist) do not conflict with each other when we apply Rubin's rules to combine these 500 separate outcome analyses as the single estimator and inference reported in Table 6. This is because the frequentist validity of applying Rubin's rules to combine separate outcome analyses with multiple imputed data sets only explicitly depends on the asymptotic normal approximation assumption for each coefficient estimator in Model (1) (see Appendix 2 for more details), and does not directly depend on how the multiple imputed data sets are generated (e.g. either using a Bayesian imputation model as in Step 3 or using a frequentist imputation model instead). Using a Bayesian imputation model followed by a frequentist substantive model is one of the most common strategies when applying Rubin's rules to conduct statistical inference with multiple imputation; see *Rubin, 1996*, Chapter 3 of *Rubin, 1987*, and Chapter 2 of *Carpenter and Kenward, 2012*. For representative works on justifying the advantages of using a Bayesian imputation model in multiple-imputation inferences, see *Meng, 1994* and Chapter 2 of *Carpenter and Kenward, 2012*.

## Secondary analyses

We also conducted the following four secondary analyses (SA1) – (SA4) which examine the causal hypothesis that reduced malaria transmission intensity cause reductions in the low birth weight rate in various ways.

- (SA1) In the first secondary analysis, we fit the mixed-effects linear probability model with multiple imputation only on the children whose age at the corresponding survey is no older than one year old (7156 out of 18,112 records) to mitigate the potential bias resulting from the births that did not occur in exactly the same year as the year of the corresponding malaria prevalence measurement.
- (SA2) In the second secondary analysis, we fit the mixed-effects linear probability model with multiple imputation over first born children only (3890 out of 18,112 records) to check if the potential effect of reduced malaria burden on the low birth weight rate is especially substantial/weak for first born children or not.
- (SA3) In the third secondary analysis, we make the difference between high malaria prevalence and low prevalence more extreme. Specifically, we redefine the malaria prevalence levels (ranging from 0 to 1) as: high ($\mathrm{PfPR}_{2-10} > 0.45$), medium ($\mathrm{PfPR}_{2-10}$ lies in [0.15, 0.45]), and low ($\mathrm{PfPR}_{2-10} < 0.15$). We then conduct the same statistical analysis procedure as in the primary analysis to check if a moderately greater reduction in malaria burden would lead to more of a decrease in the low birth weight rate or not.

- (SA4) In the fourth secondary analysis, we conduct the same procedure as in (SA3), but making the high-medium-low malaria prevalence cut-offs even more extreme: high ($\mathrm{PfPR}_{2-10} > 0.5$), medium ($\mathrm{PfPR}_{2-10}$ lies in [0.1, 0.5]), and low ($\mathrm{PfPR}_{2-10} < 0.1$) to check if a substantially more dramatic reduction in malaria burden would cause a more dramatic decrease in the low birth weight rate or not.

### Sensitivity analyses

As discussed in the 'Motivation and overview of our approach' section, using matching as a data pre-processing step in a difference-in-differences study can reduce the potential bias that may result from a violation of the parallel trend assumption arising from failing to adjust for observed covariates and the survey location sampling variability when using the survey data to conduct a difference-in-differences study. However, neither matching nor difference-in-differences can directly adjust for unobserved covariates (i.e. unmeasured confounders or events). The estimated treatment effect (i.e. the estimated coefficient of the low malaria prevalence indicator contributing to the low birth weight rate) from our primary analysis can be biased by failing to adjust for any potential unobserved covariates. How potential unobserved covariates may bias the estimated effect in a difference-in-differences study has been understood from various alternative perspectives in the previous literature. These alternative perspectives are intrinsically connected and we briefly list three of them here (for more detailed descriptions, see Appendix 3):

- Perspective 1: The potential violation of the unconfoundedness assumption (*Rosenbaum and Rubin, 1983b*; *Heckman and Robb, 1985*; *Heckman et al., 1997*).
- Perspective 2: The potential violation of the parallel trend assumption in a difference-in-differences study (*Card and Krueger, 2000*; *Angrist and Pischke, 2008*; *Hasegawa et al., 2019*; *Basu and Small, 2020*).
- Perspective 3: The difference-in-differences estimator may be biased if there is an event that is more (or less) likely to occur as the intervention happens and the occurrence probability of this event cannot be fully captured by observed covariates (*Shadish, 2010*; *West and Thoemmes, 2010*).

To assess the robustness of the results of our primary analysis to potential hidden bias, we adapt an omitted variable sensitivity analysis approach (*Rosenbaum and Rubin, 1983a*; *Imbens, 2003*; *Ichino et al., 2008*; *Zhang and Small, 2020*). Specifically, our sensitivity analysis model (i.e. Model (3) in Appendix 3) extends Model (1) by including a hypothetical unobserved covariate $U$ that is correlated with both the low malaria prevalence indicator and the low birth weight indicator. Specifically, let $U_{ij}$ denote the value of $U$ of individual $j$ in cluster $i$, we consider the following data generating process for $U_{ij}$:

$$\mathbb{P}(U_{ij} = 1) = 50\% + p_1\% \cdot \mathbb{1}(i \text{ is a low malaria prevalence cluster}) \\ + p_2\% \cdot \mathbb{1}(\text{the observed or the imputed } Y_{ij} = 1), \tag{2}$$

where $p_1$ and $p_2$ are prespecified sensitivity parameters of which the unit is a percentage point. Our sensitivity analyses investigate how the estimated treatment effect varies over a range of prespecified values for $(p_1, p_2)$. See Appendix 3 for the details of the design of the sensitivity analyses and on how our proposed sensitivity analysis model helps to address the concerns about the hidden bias from Perspectives 1–3 listed above.

## Results

In this section, we report and interpret the results of matching, primary analysis, secondary analyses, and sensitivity analyses relating changes in malaria burden to changes in the birth weight rate between 2000–2015 in sub-Saharan Africa. The R (*R Development Core Team, 2020*) code for producing all the main results and tables of this article is posted on GitHub (https://github.com/siyu-heng/Malaria-and-Low-Birth-Weight, *Heng, 2021a*, copy archived at swh:1:rev: faf6455f95bca6bab364ab95699ea7cd81af1061 *Heng, 2021b*).

## Matching

We first evaluate the performance of the first-step matching where we focus on the geographical closeness of paired early year and late year clusters from the following three perspectives: (1) the geographic proximity of the early year and the late year clusters within each pair, which is evaluated through the mean distance of two paired clusters, the within-pair longitude's correlation and latitude's correlation between the paired early year and late year clusters, and the mean values of the longitudes and the latitudes of the paired early year and late year clusters; (2) the closeness of the mean annual malaria prevalence ($PfPR_{2-10}$) of the early year and late year clusters at the early year (i.e. the early malaria prevalence year in *Table 1*); (3) the closeness of the mean annual malaria prevalence of the early year and the late year clusters at the late year (i.e. the late malaria prevalence year in *Table 1*). We report the results in *Table 4*, which indicate that the first step of our matching produced pairs of clusters which are close geographically and in their malaria prevalence at a given time. Of note, the mean Haversine distance of the early year clusters and late year clusters is 24.1 km among the 219 high-low pairs of clusters, and 28.7 km among the 219 high-high pairs of clusters. The within-pair longitudes' and latitudes' correlations between the paired early year and late year clusters among the high-low and high-high pairs are all nearly one.

We then evaluate the performance of the second-step matching, where we focus on the closeness of the sociodemographic status of paired high-low and high-high pairs of clusters, by examining the balance of each covariate among high-low and high-high pairs of early year and late year clusters before and after matching. Recall that for each cluster, we calculate the six cluster-level covariates (i.e. urban or rural, toilet facility, floor facility, electricity, mother's education level, contraception indicator) by averaging over all available individual-level records in that cluster. In each high-low or high-high pair of clusters, there are 12 associated covariates, six for the early year cluster in that pair and six for the late year cluster in that pair. *Table 5* reports the mean of each covariate among high-low pairs of clusters and high-high pairs of clusters before and after matching, along with the absolute standardized differences before and after matching. From *Table 5*, we can see that before matching, the high-high pairs are quite different from the high-low pairs, all absolute standardized differences are greater than 0.2. The high-low pairs tend to be sociodemographically better off than the high-high pairs (higher prevalence of improved toilet facilities and floor material facilities, higher prevalence of domestic electricity, higher levels of mother's education, higher rate of contraceptive use, and more urban households). To reduce the bias from these observed covariates, we leverage optimal cardinality matching, as described above, to pair a high-low pair of clusters with a high-high pair and throw away the pairs of clusters for which the associated covariates cannot be balanced

**Table 4.** The mean Haversine distance of the early year clusters and late year clusters is 24.1 km among the 219 high-low pairs of clusters, and 28.7 km among the 219 high-high pairs of clusters.

The within-pair longitudes' and latitudes' correlations between the paired early year and late year clusters among the high-low and high-high pairs all nearly equal one. The mean values of the longitudes, the latitudes, the annual malaria prevalence (i.e. $PfPR_{2-10}$) measured at the early year, denoted as $PfPR_{2-10}$ (early), and at the late year, denoted as $PfPR_{2-10}$ (late), of the paired early year clusters (clusters sampled at the early year) and late year clusters (clusters sampled at the late year) among the 219 high-low and 219 high-high pairs of clusters used for the statistical inference respectively. Note that an early year cluster has a late year $PfPR_{2-10}$ and a late year cluster has an early year $PfPR_{2-10}$ since the MAP data contain $PfPR_{2-10}$ for each location and for each year between 2000 and 2015.

| | High-low pairs | | High-high pairs | |
|---|---|---|---|---|
| Mean within-pair haversine distance | 24.1 km | | 28.7 km | |
| Within-pair correlation of longitude | 0.9999 | | 0.9996 | |
| Within-pair correlation of latitude | 0.9998 | | 0.9997 | |
| | Longitude | Latitude | $PfPR_{2-10}$ **(early)** | $PfPR_{2-10}$ **(late)** |
| Early clusters among high-low pairs | 16.92 | −1.15 | 0.52 | 0.17 |
| Late clusters among high-low pairs | 16.88 | −1.15 | 0.48 | 0.12 |
| Early clusters among high-high pairs | 19.15 | 0.43 | 0.51 | 0.47 |
| Late clusters among high-high pairs | 19.13 | 0.46 | 0.53 | 0.49 |

**Table 5.** Balance of each covariate before matching (BM) and after matching (AM).

We report the mean of each covariate (including early and late years) for high-low and high-high pairs of clusters, before and after matching. We also report each absolute standardized difference (Std.dif) before and after matching.

| | Before matching | | After matching | | Std.dif | |
|---|---|---|---|---|---|---|
| | High-low | High-high | High-low | High-high | BM | AM |
| | (410 pairs) | (540 pairs) | (219 pairs) | (219 pairs) | | |
| Urban/rural (early) | 0.44 | 0.20 | 0.26 | 0.26 | 0.53 | 0.00 |
| Urban/rural (late) | 0.60 | 0.21 | 0.37 | 0.32 | 0.85 | 0.09 |
| Toilet facility (early) | 0.88 | 0.60 | 0.82 | 0.79 | 0.86 | 0.10 |
| Toilet facility (late) | 0.94 | 0.69 | 0.90 | 0.88 | 0.90 | 0.10 |
| Floor material (early) | 1.90 | 1.68 | 1.60 | 1.67 | 0.31 | 0.10 |
| Floor material (late) | 2.22 | 1.79 | 1.92 | 1.87 | 0.59 | 0.07 |
| Electricity (early) | 0.36 | 0.12 | 0.17 | 0.16 | 0.70 | 0.02 |
| Electricity (late) | 0.54 | 0.18 | 0.33 | 0.30 | 0.99 | 0.10 |
| Mother's education (early) | 1.00 | 0.36 | 0.69 | 0.64 | 1.36 | 0.10 |
| Mother's education (late) | 1.23 | 0.42 | 0.87 | 0.83 | 1.78 | 0.10 |
| Contraception indicator (early) | 0.16 | 0.12 | 0.15 | 0.17 | 0.27 | 0.10 |
| Contraception indicator (late) | 0.22 | 0.18 | 0.24 | 0.26 | 0.23 | 0.10 |

well. After matching, we can see that all 12 covariates are balanced well – all absolute standardized differences after matching are less than 0.1.

## Effect of reduced malaria burden on the low birth weight rate

*Appendix 1—table 3* summarizes the low malaria prevalence indicators, the time indicators, the group indicators, the covariates, and the birth weights of the 18,112 births in the matched clusters. *Table 6* reports the estimated causal effect of reduced malaria burden (low vs. high malaria prevalence) on the rate of births with low birth weight, which is represented as the coefficient on the malaria prevalence indicator (diagnostics for the multiple imputation that was used in generating the estimates in *Table 6* are shown in *Appendix 2—table 1*). We estimate that a decline in malaria prevalence from $\text{PfPR}_{2-10} > 0.40$ to less than 0.20 reduces the rate of low birth weight by 1.48

**Table 6.** Inference with multiple imputation and mixed-effects linear probability model (1).

The unit of estimates and CIs is a percentage point.

| Regressor | Estimate | 95% CI | p-value |
|---|---|---|---|
| 0 - high prevalence; 1 - low prevalence | −1.48 | [−3.70, 0.74] | 0.191 |
| 0 - early year; 1 - late year | −0.06 | [−1.82, 1.69] | 0.943 |
| 0 - high-high pairs; 1 - high-low pairs | 0.21 | [−1.40, 1.82] | 0.797 |
| Mother's age (linear term) | −1.86 | [−2.48, −1.23] | <0.001*** |
| Mother's age (quadratic term) | 0.03 | [0.02, 0.04] | <0.001*** |
| Child's birth order (linear term) | −13.91 | [−18.49, −9.32] | <0.001*** |
| Child's birth order (quadratic term) | 2.91 | [1.82, 4.00] | <0.001*** |
| Wealth index | 0.09 | [−0.38, 0.56] | 0.709 |
| 0 - rural; 1 - urban | 0.82 | [−0.63, 2.27] | 0.269 |
| Mother's education level | −2.02 | [−2.82, −1.22] | <0.001*** |
| Child is boy | −1.75 | [−2.75, −0.74] | <0.001*** |
| Mother is married or living together | −1.43 | [−3.04, 0.19] | 0.083 |
| Antenatal care indicator | −0.96 | [−2.06, 0.13] | 0.085 |

percentage points (95% confidence interval: 3.70 percentage points reduction, 0.74 percentage points increase). A reduction in the low birth weight rate of 1.48 percentage points is substantial; recall that among the study individuals with nonmissing birth weight, the low birth weight rate was 8.6%, so a 1.48 percentage points reduction corresponds to a 17% reduction in the low birth weight rate. The results in *Table 6* also show that there is strong evidence that mother's age, child's birth order, mother's education level and child's sex are also associated with the low birth weight rate. For example, mothers with higher education level are less likely to deliver a child with low birth weight, and boys are less likely to have low birth weight than girls, which agrees with the previous literature (e.g. *Brooke et al., 1989*; *Valero De Bernabé et al., 2004*; *Zeka et al., 2008*). Our estimated reduction in the low birth weight rate of 1.48 percentage points from reducing malaria prevalence from high to low is similar to that from a naive difference-in-differences estimator that ignores covariates and missingness of birth weight records. The observed low birth weight rates among the records with observed birth weight within the early year clusters in high-low pairs is 9.33%, in the late year clusters in high-low pairs is 7.52%, in the early year clusters in high-high pairs is 9.18%, and in the late year clusters in high-high pairs is 9.06%. Therefore, the naive difference-in-differences estimator for the effect of reduced malaria burden without adjusting for covariates and missingness of birth weight records is $(7.52\% - 9.33\%) - (9.06\% - 9.18\%) = -1.69\%$ (i.e. 1.69 percentage points reduction on the low birth weight rate).

Among all the high-low pairs of clusters in our sample, there has been a decrease in the low birth weight rate from the early years to the late years of 1.81 percentage points (from 9.33% to 7.52%) for records with observed birth weight and an estimated decrease of 2.04 percentage points (from 10.48% to 8.44%) when multiple imputation is used to impute missing birth weight records. We now explore how much of this decrease can be attributed to reduced malaria burden over time. The estimated effect in *Table 6* of the time indicator (late year vs. early year) is a 0.06 percentage points reduction, which is much less than that of the low malaria prevalence indicator. Moreover, the estimated change in the low birth weight rate over time among high-low pairs that comes from changes in the covariates over time is a 0.52 percentage points reduction. This is calculated by looking at the difference between $\widehat{\beta}^T \overline{\mathbf{x}}_{\text{early}}$ and $\widehat{\beta}^T \overline{\mathbf{x}}_{\text{late}}$, where $\widehat{\beta}^T$ is the estimated coefficients of the covariate regressors listed in *Table 6*, and $\overline{\mathbf{x}}_{\text{early}}$ and $\overline{\mathbf{x}}_{\text{late}}$ are the average values in high-low pairs of the covariate regressors of the individuals within the early year clusters and those within the late year clusters respectively. These results suggest that after adjusting for the observed covariates listed in *Table 6* and missingness of birth weight records, the observed decrease in the low birth weight rate over time in high-low pairs comes mainly from reduced malaria burden over time instead of changes over time in the low birth weight rate that affect both high-low and high-high pairs of clusters. To illustrate this point and further verify the potentially substantial effect of reduced malaria burden on the low birth weight rate, we also plot the estimated low birth weight rate of each cluster among the high-high pairs and high-low pairs in our study sample in *Figure 3*. From *Figure 3*, we can see that although in general, for both high-high pairs and high-low pairs, the birth weight rates of the late year clusters are lower than those of the early year clusters, it is clear that the reductions in low birth weight rate from early year to late year among the high-low pairs are considerably greater than those among high-high pairs, suggesting that reducing community-level malaria burden can potentially substantially reduce the low birth weight rate.

## Results of secondary analyses

The results of our secondary analyses support the interpretation of our primary analysis:

- (SA1) In the first secondary analysis, when only conducting statistical analysis among children whose age at the survey year is no older than 1 year, the point estimate of the coefficient of the low malaria prevalence indicator (1 if $\text{PfPR}_{2-10} < 0.2$) is $-1.31$ percentage points (95% CI: $[-4.70, 2.08]$), which in general agrees with the result of our primary analysis and implies that our causal conclusion drawn from the primary analysis is relatively robust to the potential hidden bias caused by the births that occurred in different years from the years of the malaria prevalence measurement.
- (SA2) In the second secondary analysis, performing our statistical analysis among first born children only, the estimated coefficient of the low malaria prevalence indicator is $-3.73$ percentage points (95% CI: $[-9.11, 1.64]$). This implies that the effect of reduced malaria burden on the low birth weight rate may be especially substantial among first born children.

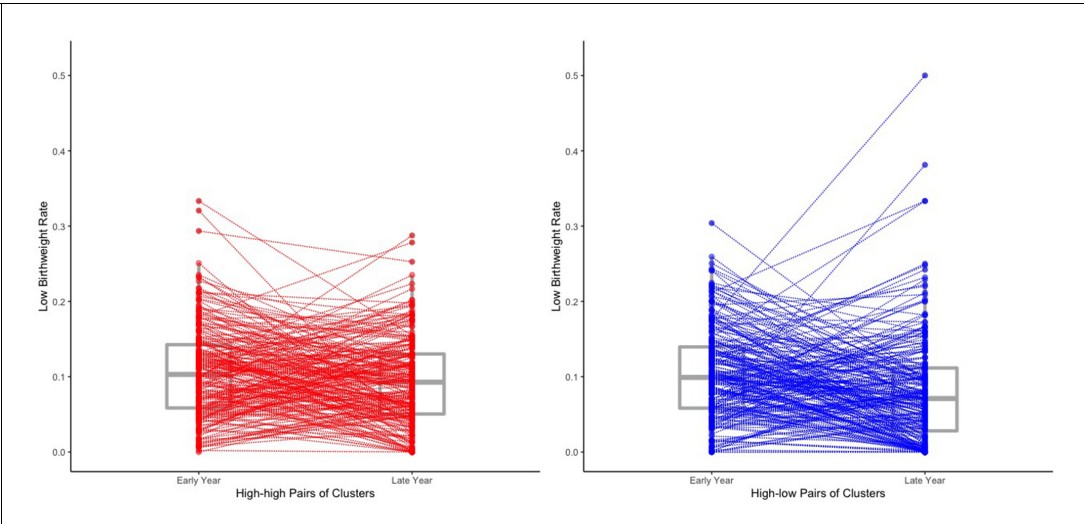

**Figure 3.** The estimated low birth weight rate of each cluster within the 219 high-high pairs and the 219 high-low pairs. The estimated low birth weight rate for each cluster are obtained from averaging over all the 500 imputed data sets of the 18,112 individual records. We draw a line to connect two paired clusters (one early year cluster and one late year cluster). Box plots for the low birth weight rates are also shown. Two of the four outliers of the late year clusters among the high-low pairs (i.e. the top four late year clusters in terms of low birth weight rate among the high-low pairs) may result from their extremely small within-cluster sample sizes (no more than three individual records for both two clusters).

- (SA3) In the third secondary analysis, after slightly enlarging the difference between high malaria prevalence and low prevalence and repeating the two-stage matching procedure described above, there remain 100 high-high pairs of clusters and 100 high-low pairs, with 8611 individual records remaining in the final model. In (SA3), the point estimate of the coefficient of low malaria prevalence indicator is $-1.48$ percentage points (95% CI: $[-4.44, 1.48]$). In this case, slightly enlarging the gap between the cutoffs for high/low malaria prevalence did not result in an obvious additional reduction in the low birth weight rate. A possible reason is that the new cut-offs are just slightly different from the previous ones and the changes may still lie within the margin of error of measuring the $\mathrm{PfPR}_{2-10}$ or there may not be enough power. In thinking about the results of (SA3), it is useful to also consider the results from (SA4).
- (SA4) In the fourth secondary analysis, after making the high prevalence and low prevalence cut-offs quite extreme and repeating the two-stage matching procedure, there remain 35 high-high pairs of clusters and 35 high-low pairs, with 3135 individual records remaining in the final model. In (SA4), the point estimate of the coefficient of low malaria prevalence indicator is $-3.04$ percentage points (95% CI: $[-8.50, 2.41]$). This implies that a more dramatic reduction in malaria burden can potentially lead to a more dramatic decrease in the low birth weight rate and supports the above hypothesis that the fact that slightly enlarging the gap between the high/low malaria prevalence cutoffs in (SA3) did not result in an evident additional reduction in the low birth weight rate may be due to the potential measurement error of the $\mathrm{PfPR}_{2-10}$ or lack of power.

### Results of the sensitivity analyses

Recall that in the 'Sensitivity analyses' section and Appendix 3, our sensitivity analyses consider a hypothetical unobserved covariate $U$ that is correlated with both the low malaria prevalence indicator and the low birth weight indicator. For various values of the sensitivity parameters $(p_1, p_2)$, we report the corresponding point estimates and 95% CIs of the estimated treatment effect (i.e. the coefficient of the low malaria prevalence indicator contributing to the low birth weight rate) in *Appendix 3—table 1*. The results from *Appendix 3—table 1* show that the estimated treatment effect ranges from 1.13 percentage points reduction to 1.83 percentage points reduction (on the low birth weight rate) if both $p_1$ and $p_2$ are between $-10$ and 10. Recall that $p_1$ (or $p_2$) equals 10 (or $-10$) means that the probability of the $U$ taking value one increases (or decreases) by 10 percentage points if the individual's low malaria prevalence indicator (or the low birth weight rate indicator) equals 1. That is, allowing both the magnitude of $p_1$ and the magnitude of $p_2$ can be up to 10 means

that we allow the existence of a nontrivial magnitude of unmeasured confounding in our sensitivity analyses. Therefore, the estimated treatment effect ranging from 1.13 percentage points reduction to 1.83 percentage points reduction when both $p_1$ and $p_2$ are between $-10$ and $10$ means that the magnitude of the estimated treatment effect is still evident (no less than 1.13 percentage points) even if the magnitude of unmeasured confounding is nontrivial (both $|p_1|$ and $|p_2|$ can be up to 10). See Appendix 3 for the detailed results and interpretations of the sensitivity analyses.

To conclude, although the confidence intervals of the coefficient of the low malaria prevalence indicator on the low birth weight rate presented in the 'Results' section cannot exclude a possibility of no effect at level 95% based on our proposed study sample selection procedure and statistical approach, the results and the corresponding interpretations of the primary analysis, the secondary analyses, and the sensitivity analyses have contributed to the weight of the evidence that reduced malaria burden has an important influence on the low birth weight rate in sub-Saharan Africa at the community level.

## Discussion

We have developed a pair-of-pairs matching approach to conduct a difference-in-differences study to examine the causal effect of a reduction in malaria prevalence on the low birth weight rate in sub-Saharan Africa during the years 2000–2015. Although we cannot rule out no effect at a 95% confidence level, the magnitude of the estimated effect of a reduction from high malaria prevalence to low malaria prevalence on the low birth weight rate (1.48 percentage points) is even greater than the estimated effect of a factor thought to be important, antenatal care during pregnancy (0.96 percentage points). In a secondary analysis, we find that reduction in malaria burden from high to low is estimated to be especially crucial for reducing the low birth weight rate of first born children, reducing it by 3.73 percentage points (95% CI: 9.11 percentage points reduction, 1.64 percentage points increase). This agrees with previous studies which demonstrate that the effects of malaria on birth outcomes are most pronounced in the first pregnancy (e.g. *McGregor et al., 1983*).

Previous studies have shown that individual malaria prevention during pregnancy reduces the chances of the woman's baby having low birth weight (*Kayentao et al., 2013*). In this paper, we examine the community-level effect of reductions in malaria on pregnancy outcomes as opposed to the individual-level effect of malaria prevention interventions during pregnancy. Our results support extrapolation of studies of antenatal malaria interventions on birth weight to populations experiencing declining malaria burden. Furthermore, we conclude that reports of declining malaria mortality underestimate the contribution of reduced malaria exposure during pregnancy on pregnancy outcomes and neonatal survival. Although some studies have documented higher rates of adverse pregnancy outcomes in malaria-infected women with declining antimalarial immunity, such as may be seen in communities with declining malaria exposure (*Mayor et al., 2015*), our study demonstrates that overall reduction in exposure to infection, including during pregnancy, outweighs these individual changes in risk once infected.

Strengths of our study include that we use state-of-the-art causal inference methods on a large representative data set. We develop a novel pair-of-pairs matching approach to conduct a difference-in-differences study to estimate the real world effectiveness of public health interventions by combining DHS data with other data sources. There are two major difficulties when using the DHS data to conduct a difference-in-differences study. The first major difficulty is that within each country the DHS samples different locations (clusters) over different survey years. Our first-step matching handles this difficulty through using optimal matching to pair the early year DHS clusters and the late year DHS clusters within the same country based on the geographic proximity of their locations. Then each formed pair of clusters can mimic a single cluster measured twice in two different survey years, which serves as the foundation of a difference-in-differences study. The second major difficulty is that although an advantage of the DHS data is that they contain many potentially important cluster-level and individual-level covariates, it may be difficult to come up with a statistical model that is both efficient and robust to adjust for these covariates. A traditional approach to estimating the real world effectiveness of an intervention in such settings is to run a regression of an outcome of interest on a measure of adherence to the treatment (zero if in the period before the intervention was available and ranging from 0 to 1 after the intervention was available), covariates (individual-level and cluster-level covariates) and a random effect for the cluster (*Goetgeluk and Vansteelandt, 2008*).

This regression approach relies heavily on correct specification of the model by which the covariates affect the outcome (e.g. linear, quadratic, cubic), therefore the result can be severely biased by model misspecification (*Rubin, 1973*; *Hansen, 2004*; *Ho et al., 2007*). We instead use a second-step matching to first optimally select and match the treated units (i.e. high-low pairs of clusters) and control units (i.e. high-high pairs of clusters) to ensure that they have balanced distributions of covariates across time and then run the regression with the dummy variables for the matched sets. Such a nonparametric data preprocessing step before running a regression can potentially reduce bias due to model misspecification (*Rubin, 1973*; *Hansen, 2004*; *Ho et al., 2007*).

Our merged study data set makes use of two aspects of the richness of the relevant data resources. First, from the perspective of sample size and length of time span, the data set includes over 18,000 births in 19 countries in sub-Saharan Africa and describes changes in the low birth weight rate over a 15-year period. Some of the studied regions had substantial changes in malaria parasite prevalence during this time period, whereas others did not, which provides us ample heterogeneity necessary for conducting a difference-in-differences study. Second, from the perspective of the comprehensiveness of information, our merged data set includes various types of information: from cluster-level to individual-level records; from geographic to sociodemographic characteristics; from surveyed data to predicted data.

Some potential limitations of our study should be considered. First, we discretized the mean malaria prevalence (i.e. $\mathrm{PfPR}_{2-10}$ from 0 to 1) into high ($\mathrm{PfPR}_{2-10} > 0.4$), medium ($\mathrm{PfPR}_{2-10}$ lies in [0.2, 0.4]), and low ($\mathrm{PfPR}_{2-10} < 0.2$), which means that the magnitude of the estimated causal effect depends on how we define these cut-offs. Our primary analysis suggests that reducing the malaria burden from high to low may substantially help control the low birth weight rate, and our secondary analyses suggest that a more dramatic reduction in malaria prevalence can lead to a more dramatic drop in the low birth weight rate. More research needs to be done on the minimum magnitude of the reduction in malaria prevalence that is needed to cause a substantial drop in the low birth weight rate. Second, we assigned the malaria prevalence (i.e. $\mathrm{PfPR}_{2-10}$) data to children's records based on the DHS survey years which may not be exactly the same years as children's actual birth years. For example, a child whose age is three years at the corresponding DHS survey year should have been born three years earlier before that DHS survey year, in which case we might have assigned the wrong $\mathrm{PfPR}_{2-10}$ to that child's gestational period. We examined this issue via SA1 and the result suggested that this did not induce much bias to the results of our primary analysis.

The novel design-based causal inference approach developed in this work, a pair-of-pairs matching approach to conduct a difference-in-differences study (i.e. the two-step matching procedure to form matched pairs of pairs as a nonparametric data preprocessing step in a difference-in-differences study), is potentially useful for researchers who would like to reduce the estimation bias due to potential model misspecification in the traditional difference-in-differences approach. Moreover, the general statistical methodology developed in this work can be applied beyond the malaria settings to handle the heterogeneity of survey time points and locations in data sets such as the Demographic and Health Surveys (DHS).

In summary, the contribution of malaria to stillbirth and neonatal mortality, for which low birth weight is a proxy, are currently not accounted for in global estimates of malaria mortality. Using a large representative data set and innovative statistical evidence, we found point estimates that suggested that reductions in malaria burden at the community level substantially reduce the low birth weight rate. To our knowledge, this is the first study of its kind to evaluate the causal effects of malaria control on birth outcomes using a causal inference framework. Although our confidence intervals do include a possibility of no effect, the evidence from our primary analysis and secondary analyses is strong enough to merit further study and motivate further investments in mitigating the intolerable burden of malaria.

## Acknowledgements

The authors thank Bhaswar Bhattacharya, Shuxiao Chen, Emily Diana, Sheng Gao, Bikram Karmakar, Hongming Pu, Hua Wang, and Bo Zhang for helpful discussions and comments. Funding: Ryan A Simmons is supported by National Center for Advancing Translational Sciences (UL1TR002553). The funder had no role in study design, data collection and interpretation, or the decision to submit the work for publication.

## Additional information

### Funding

| Funder | Grant reference number | Author |
| --- | --- | --- |
| National Center for Advancing Translational Sciences | UL1TR002553 | Ryan Simmons |

The funders had no role in study design, data collection and interpretation, or the decision to submit the work for publication.

### Author contributions

Siyu Heng, Data curation, Software, Formal analysis, Investigation, Visualization, Methodology, Writing - original draft; Wendy P O'Meara, Conceptualization, Resources, Data curation, Supervision, Investigation, Methodology, Writing - review and editing; Ryan A Simmons, Resources, Data curation, Investigation, Methodology, Writing - review and editing; Dylan S Small, Conceptualization, Resources, Supervision, Investigation, Methodology, Writing - review and editing

### Author ORCIDs

Siyu Heng (iD) https://orcid.org/0000-0002-9313-3667
Dylan S Small (iD) https://orcid.org/0000-0003-4928-2646

### Decision letter and Author response

Decision letter https://doi.org/10.7554/eLife.65133.sa1
Author response https://doi.org/10.7554/eLife.65133.sa2

## Additional files

### Supplementary files

• Source code 1. The source code for producing the results in *Figure 1*, the results in *Figure 3*, the results in *Tables 2*, *3*, *5* and *6*, the results in *Table 4*, the results in *Appendix 1—table 2*, the results in Appendix 1 *Table 3*, and the results in *Appendix 3—table 1* can be found respectively in 'Code for Figure 1.R', 'Code for Figure 3.R', 'Code for primary analysis.R', 'Code for Table 4.R', 'Code for Appendix 1 Table 2.R', 'Code for Appendix 1 Table 3.R', and 'Code for Sensitivity Analyses.R' in the source code files. The source code are also posted on GitHub (https://github.com/siyuheng/Malaria-and-Low-Birth-Weight; *Heng, 2021a*).

• Transparent reporting form

### Data availability

Our study is an observational study that uses the following three publicly available data sets: (1) Annual Mean of Plasmodium falciparum Parasite Rate, created by the Malaria Atlas Project, available at https://malariaatlas.org/malaria-burden-data-download/; (2) Integrated Public Use Microdata Series' recoding of the DHS variables (IPUMS-DHS), created by the IPUMS-DHS Team, available at https://www.idhsdata.org/idhs/index.shtml; (3) DHS-GPS data, created by the Demographic and Health Surveys (DHS) Program, available at https://dhsprogram.com/data/available-datasets.cfm. The data resources are also described in the "Data resources" section of the manuscript. The source code for producing the results in Figure 1, the results in Figure 3, the results in Tables 2, 3, 5, and 6, the results in Table 4, the results in Appendix 1—table 2, the results in Appendix 1—table 3, and the results in Appendix 3 —table 1 can be found in Source code 1 and on GitHub at https://github.com/siyuheng/Malaria-and-Low-Birth-Weight (copy archived at https://archive.softwareheritage.org/swh:1:rev:faf6455f95bca6bab364ab95699ea7cd81af1061).

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

# Appendix 1

## More details on the data selection procedure

We give more details on how we select the study countries (among all sub-Saharan African countries) and their corresponding late year and early year for each of the three data sets: malaria prevalence data (MAP data), IPUMS-DHS data, and DHS cluster GPS data. We define 'early year' as 2000–2007 and 'late year' as 2008–2015. We first select countries that have both IPUMS-DHS data and DHS GPS data for at least one year between 2000–2007 and one year between 2008–2015. If there are more than one early (late) years available, we choose the earliest early year and latest late year. Note that some DHS can span over two years. In this case, we stick to the way how IPUMS-DHS codes the year of that DHS data set. For example, both Malawi and Tanzania have a standard DHS with GPS data that spans over 2015–2016. We call them Malawi 2015–2016 DHS and Tanzania 2015–2016 DHS respectively. In IPUMS-DHS, the year for Malawi 2015–2016 DHS is coded as 2016, and that for Tanzania 2015–2016 DHS is coded as 2015. Therefore, for Malawi, we use Malawi 2010 DHS as the study sample for the late year and exclude Malawi 2015–2016 DHS. While for Tanzania, we use Tanzania 2015 DHS for the late year. As we have mentioned in the main text, if a country has at least one year between 2008–2015 with available IPUMS-DHS data of which the GPS data is also available, but no available IPUMS-DHS data or the corresponding GPS data between 2000–2007, we still include that country if it has IPUMS-DHS data along with the corresponding GPS data for the year 1999 (possibly with overlap into 1998). This selection procedure results in 19 study countries in total. Note that for the DHS that span over two successive years, sometimes IPUMS-DHS and the GPS data code their years in different ways. In these cases, when attaching the malaria prevalence data to each cluster, we stick to the year which is used by the GPS data; see *Appendix 1—table 1*. For example, for Benin 2011–2012 DHS, IPUMS-DHS codes its year as 2011 while the GPS data codes its year as 2012. In these cases, we use the malaria prevalence data for 2012 for the clusters within Benin 2011–2012; see the row 'Benin (BJ)' in *Appendix 1—table 1*.

**Appendix 1—table 1.** The early and late years coded in the IPUMS-DHS and GPS data sets.

|  | GPS data | | Malaria prevalence | | IPUMS-DHS | |
| --- | --- | --- | --- | --- | --- | --- |
|  | Early | Late | Early | Late | Early | Late |
| Benin | 2001 | 2012 | 2001 | 2012 | 2001 | 2011 |
| Burkina Faso (BF) | 2003 | 2010 | 2003 | 2010 | 2003 | 2010 |
| Cameron (CM) | 2004 | 2011 | 2004 | 2011 | 2004 | 2011 |
| Congo Democratic Republic (CD) | 2007 | 2013 | 2007 | 2013 | 2007 | 2013 |
| Cote d'Ivoire (CI) | 1998 | 2012 | 2000 | 2012 | 1998 | 2011 |
| Ethiopia (ET) | 2000 | 2010 | 2000 | 2010 | 2000 | 2011 |
| Ghana (GH) | 2003 | 2014 | 2003 | 2014 | 2003 | 2014 |
| Guinea (GN) | 2005 | 2012 | 2005 | 2012 | 2005 | 2012 |
| Kenya (KE) | 2003 | 2014 | 2003 | 2014 | 2003 | 2014 |
| Malawi (MW) | 2000 | 2010 | 2000 | 2010 | 2000 | 2010 |
| Mali (ML) | 2001 | 2012 | 2001 | 2012 | 2001 | 2012 |
| Namibia (NM) | 2000 | 2013 | 2000 | 2013 | 2000 | 2013 |
| Nigeria (NG) | 2003 | 2013 | 2003 | 2013 | 2003 | 2013 |
| Rwanda (RW) | 2005 | 2014 | 2005 | 2014 | 2005 | 2014 |
| Senegal (SN) | 2005 | 2010 | 2005 | 2010 | 2005 | 2010 |
| Tanzania (TZ) | 1999 | 2015 | 2000 | 2015 | 1999 | 2015 |
| Uganda (UG) | 2000 | 2011 | 2000 | 2011 | 2001 | 2011 |
| Zambia (ZM) | 2007 | 2013 | 2007 | 2013 | 2007 | 2013 |
| Zimbabwe (ZW) | 2005 | 2015 | 2005 | 2015 | 2005 | 2015 |

## Country summary

**Appendix 1—table 2.** The numbers of the high-high pairs of clusters and high-low pairs of clusters contributed by each of the 19 selected sub-Saharan African countries after the matching in Step 1 and Step 2.
We also summarize the total number of pairs of clusters after Step 1 matching in the first column.

| Country | Step 1 matching | | | Step 2 matching | |
|---|---|---|---|---|---|
| | Total pairs | High-high | High-low | High-high | High-low |
| Benin | 247 | 29 | 6 | 4 | 6 |
| Burkina Faso | 400 | 150 | 0 | 19 | 0 |
| Cameron | 466 | 17 | 163 | 16 | 51 |
| Congo Democratic Republic | 300 | 11 | 55 | 11 | 24 |
| Cote d'Ivoire | 140 | 19 | 2 | 7 | 2 |
| Ethiopia | 539 | 0 | 0 | 0 | 0 |
| Ghana | 412 | 24 | 18 | 18 | 8 |
| Guinea | 295 | 47 | 12 | 10 | 12 |
| Kenya | 400 | 2 | 10 | 2 | 8 |
| Malawi | 560 | 96 | 15 | 81 | 15 |
| Mali | 402 | 101 | 21 | 17 | 19 |
| Namibia | 260 | 0 | 0 | 0 | 0 |
| Nigeria | 362 | 24 | 11 | 16 | 1 |
| Rwanda | 462 | 0 | 0 | 0 | 0 |
| Senegal | 376 | 0 | 0 | 0 | 0 |
| Tanzania | 176 | 0 | 68 | 0 | 57 |
| Uganda | 298 | 19 | 29 | 17 | 16 |
| Zambia | 319 | 1 | 0 | 1 | 0 |
| Zimbabwe | 398 | 0 | 0 | 0 | 0 |
| Total | 6812 | 540 | 410 | 219 | 219 |

## Some remarks on the IPUMS-DHS data used in this article

There are different units of analysis for data browsing in IPUMS-DHS (*Boyle et al., 2019*). In 'Step 2: Matching on sociodemographic similarity is emphasized in second matching,' for the covariates 'Household electricity,' 'Household main material of floor,' and 'Household toilet facility,' the IPUMS-DHS data we used is at the household members level (each record is a household member). For the covariates 'Mother's education level' and 'Indicator of whether the woman is currently using a modern method of contraception,' the IPUMS-DHS data we used is at the birth level (each record is a birth reported by a woman of childbearing age). The covariate 'Urban or rural' obtained from the DHS GPS data is at the DHS clusters level. In 'Step 3: Low birth weight indicator with multiple imputation to address missingness' and 'Step 4: Estimation of causal effect of reduced malaria burden on the low birth weight rate,' the IPUMS-DHS data we used is at the child level (each record is a child under age 5).

# More details on the final study population

**Appendix 1—table 3.** Summary of the low malaria prevalence indicators, the time indicators, the group indicators, the covariates, and the birth weight records among the 18,112 study individual records.

| Variables | Percentages of some categories |
| --- | --- |
| Low malaria prevalence indicator | High prevalence (70.6%) |
| | Low prevalence (29.4%) |
| Time indicator | Early year (50.3%) |
| | Late year (49.7%) |
| Group indicator | High-high pairs (40.9%) |
| | High-low pairs (59.1%) |
| Mother's age in years | ≤19 (7.1%) |
| | 20–29 (52.5%) |
| | 30–39 (31.4%) |
| | ≥40 (8.9%) |
| Wealth index | Poorest (20.2%) |
| | Poorer (23.3%) |
| | Middle (22.8%) |
| | Richer (20.4%) |
| | Richest (13.3%) |
| Child's birth order | 1 (21.5%) |
| | 2–4 (46.0%) |
| | 4+ (32.6%) |
| Urban or rural | Rural (77.1%) |
| | Urban (22.9%) |
| Mother's education level | No education (36.6%) |
| | Primary (47.2%) |
| | Secondary or higher (16.2%) |
| Child's sex | Female (49.3%) |
| | Male (50.7%) |
| Mother's marital status | Never married or formerly in union (11.6%) |
| | Married or living together (88.4%) |
| Indicator of antenatal care | Yes (61.9%) |
| | No or missing (38.1%) |
| Self-reported birth size | Very small or smaller than average (13.0%) |
| | Average (45.5%) |
| | Larger than average or very large (41.5%) |
| Low birth weight indicator | Yes (4.6%) |
| | No (48.5%) or Missing (47.0%) |

## Appendix 2

### Statistical inference with multiple imputation applying Rubin's rules

We apply Rubin's rules (**Rubin, 1987**; **Schafer, 1999**; **Carpenter and Kenward, 2012**) to combine all the imputed data sets to obtain the point estimate, the p-value, and the 95% confidence interval for each coefficient in the mixed-effects linear probability model (1) summarized in **Table 6** of the main text. Suppose that there are $M$ imputed data sets ($M = 500$ in our study). Suppose that for the $m$-th imputed data set, $m = 1, \ldots, 500$, the estimate for the coefficient of the $i$-th regressor $\gamma_i$ (including the intercept term), $i = 1, \ldots, 14$, is $\widehat{\gamma}_{m,i}$, and let $V_i$ be its squared standard error and $\widehat{V}_{m,i}$ be the estimated squared standard error from the $m$-th imputed data set. Suppose that the following normal approximations hold

$$(\widehat{\gamma}_{m,i} - \gamma_i)/\sqrt{\widehat{V}_{m,i}} \sim \mathcal{N}(0,1), \quad i = 1, \ldots, 14, \quad m = 1, \ldots, 500.$$

According to Rubin's rules (**Rubin, 1987**; **Schafer, 1999**; **Carpenter and Kenward, 2012**), we estimate $\gamma_i$ with $\overline{\gamma}_i = M^{-1} \sum_{m=1}^{M} \widehat{\gamma}_{m,i}$. Consider the corresponding between-imputation variance $B_i = (M-1)^{-1} \sum_{m=1}^{M} (\widehat{\gamma}_{m,i} - \overline{\gamma}_i)^2$ and the within-imputation variance $\overline{V}_i = M^{-1} \sum_{m=1}^{M} \widehat{V}_{m,i}$. Then the estimated total variance is

$$T_i = (1 + M^{-1})B_i + \overline{V}_i, \quad i = 1, \ldots, 14.$$

Then we can get the corresponding two-sided p-values and 95% confidence intervals based on a Student's $t$-approximation

$(\overline{\gamma}_i - \gamma_i)/\sqrt{T_i} \sim t_{v_i}, \quad i = 1, \ldots, 14$, with degrees of freedom $v_i = (m-1)[1 + \frac{\overline{V}_i}{(1+M^{-1})B_i}]^2$.

### Multiple imputation diagnostics

Note that in our multiple imputation procedure, the variance ratios are all less than 0.5, indicating that for each regressor the variance due to missing data (between-imputation variance) is much less than the average estimated squared standard error over the 500 imputed data sets. More replications of imputation (larger $M$) will more sufficiently reduce the variation due to missingness and therefore lead to more reliable estimation (**Rubin, 1987**; **Schafer, 1999**). We take a sufficiently large number of replications $M = 500$ to ensure that the variance due to missingness has been sufficiently controlled.

**Appendix 2—table 1.** Diagnostics for multiple imputation with the mixed-effects linear probability model.
We report the between-imputation variance ('Between var'), the within-imputation variance ('Within var'), and the variance ratio: (between-imputation variance)/(within-imputation variance), denoted as 'Var ratio'.

| Regressor | Between var | Within var | Var ratio |
|---|---|---|---|
| 0 - high prevalence; 1 - low prevalence | $3.21 \times 10^{-5}$ | $9.62 \times 10^{-5}$ | 0.334 |
| 0 - early year; 1 - late year | $2.20 \times 10^{-5}$ | $5.81 \times 10^{-5}$ | 0.379 |
| 0 - high-high pairs; 1 - high-low pairs | $1.92 \times 10^{-5}$ | $4.83 \times 10^{-5}$ | 0.398 |
| Mother's age (linear term) | $3.32 \times 10^{-6}$ | $6.85 \times 10^{-6}$ | 0.486 |
| Mother's age (quadratic term) | $8.28 \times 10^{-10}$ | $1.68 \times 10^{-9}$ | 0.493 |
| Child's birth order (linear term) | $1.60 \times 10^{-4}$ | $3.87 \times 10^{-4}$ | 0.413 |
| Child's birth order (quadratic term) | $8.55 \times 10^{-6}$ | $2.24 \times 10^{-5}$ | 0.382 |
| Wealth index | $1.74 \times 10^{-6}$ | $4.05 \times 10^{-6}$ | 0.430 |
| 0 -rural; 1 - urban | $1.27 \times 10^{-5}$ | $4.21 \times 10^{-5}$ | 0.303 |

*Continued on next page*

*Appendix 2—table 1 continued*

| Regressor | Between var | Within var | Var ratio |
|---|---|---|---|
| Mother's education level | $4.56 \times 10^{-6}$ | $1.20 \times 10^{-5}$ | 0.380 |
| Child is boy | $7.12 \times 10^{-6}$ | $1.91 \times 10^{-5}$ | 0.373 |
| Mother is married or living together | $1.83 \times 10^{-5}$ | $4.96 \times 10^{-5}$ | 0.370 |
| Antenatal care indicator | $9.63 \times 10^{-6}$ | $2.16 \times 10^{-5}$ | 0.447 |

## Appendix 3

### Design of the sensitivity analyses

In Section 'Sensitivity analyses' of the main text, we very briefly described three perspectives on how potential unobserved covariates that cannot be adjusted by matching may bias the estimated effect in a difference-in-differences study. Here we give more detailed descriptions of them with connections to our study for reference:

- Perspective 1: The potential violation of the unconfoundedness assumption (*Rosenbaum and Rubin, 1983b*; *Heckman and Robb, 1985*). Roughly speaking, the unconfoundedness assumption states that, after adjusting for observed covariates (measured confounders), there are no differential trends over time of any characteristics, other than the intervention itself, between the treated group and the control group, that may be correlated with their outcomes. This assumption may be violated if there is selection bias on unobserved covariates across time (*Heckman and Robb, 1985*; *Heckman et al., 1997*) such that there are differences in these observed covariates of the treated group and the control group which impact their trends in the outcome (*Ashenfelter and Card, 1985*; *Doyle et al., 2018*). For example, in our study, the unconfoundedness assumption can be violated if the sharp drops in malaria prevalence experienced by some areas could be explained by the changes of some unobserved characteristics over time that could also predict the low birth weight rate.
- Perspective 2: The potential violation of the parallel trend assumption in a difference-in-differences study (*Card and Krueger, 2000*; *Angrist and Pischke, 2008*; *Hasegawa et al., 2019*; *Basu and Small, 2020*). Recall that the parallel trend assumption behind a difference-in-differences study states that, in the absence of the treatment (i.e. intervention), after adjusting for relevant covariates, the outcome trajectory of the treated group would follow a parallel trend with that of the control group. Therefore, to make the parallel trend assumption more likely to hold, ideally each observed or unobserved covariate should be well balanced (i.e. follow a common trajectory) between the treated group and the control group, before and after the intervention. Matching can balance observed covariates by ensuring each covariate follows a common trajectory in the treated and control groups. However, matching cannot directly adjust for unobserved covariates and their trajectories among the treated and control groups may differ and correspondingly the parallel trend assumption may not hold.
- Perspective 3: A difference-in-differences study may be biased if there is an event that is more (or less) likely to occur as the treatment (i.e. intervention) happens in the treated group, but, unlike the case discussed in Section 'Motivation and overview of our approach' of the main text, the occurrence probability of this event cannot be fully captured by observed covariates. In this case, if this event can affect the outcome, its contribution to the outcome will be more (or less) substantial within the treated group after the treatment (i.e. intervention) than that within the control group (*Shadish, 2010*; *West and Thoemmes, 2010*). For example, areas experiencing sharp drops in malaria prevalence might also be more likely to experience other events (e.g. sharp drops in the prevalence of other infectious diseases) that can contribute to decreasing the low birth weight rate.

We use an omitted variable sensitivity analysis approach (*Rosenbaum and Rubin, 1983a*; *Imbens, 2003*; *Ichino et al., 2008*; *Zhang and Small, 2020*) to evaluate the sensitivity of the results of our primary analysis to potential hidden bias caused by unobserved covariates. Specifically, we propose the following sensitivity analysis model (3) which extends Model (1) by considering a hypothetical unobserved covariate (unmeasured confounding variable or event) $U$ that is correlated with both the low malaria prevalence indicator and the low birth weight indicator. Let $U_{ij}$ denote the exact value of $U$ for individual $j$ in cluster $i$, we consider:

$$\mathbb{P}(Y_{ij} = 1 \mid i, \mathbf{X}_{ij}, U_{ij}) = k_0 + k_1 \cdot \mathbb{1}(i \text{ is a low malaria prevalence cluster}) + k_2 \cdot \mathbb{1}(i \text{ is a late year cluster})$$
$$+ k_3 \cdot \mathbb{1}(i \text{ is from a high-low pair of clusters}) + \beta^T \mathbf{X}_{ij} + \lambda \cdot U_{ij},$$

(3)

with two error terms $\alpha_i \sim \mathcal{N}(0, \sigma_0)$ and $\epsilon_{ij} \sim \mathcal{N}(0, \sigma_1)$. We assume that $U_{ij}$ follows a Bernoulli distribution (taking value 0 or 1) with

$$\mathbb{P}(U_{ij} = 1) = 50\% + p_1\% \cdot \mathbb{1}(i \text{ is a low malaria prevalence cluster})$$
$$+ p_2\% \cdot \mathbb{1}(\text{the observed or the imputed } Y_{ij} = 1).$$

(4)

In Model (3) along with the corresponding data generating model (4) of the unobserved covariate $U$, the $(p_1, p_2)$ are sensitivity parameters of which the unit is a percentage point. Prespecifying a positive (or negative) $p_1$ corresponds to a positive (or negative) correlation between the unobserved covariate $U$ and the low malaria prevalence indicator, and prespecifying a positive (or negative) $p_2$ corresponds to a positive (or negative) correlation between the unobserved covariate $U$ and the low birth weight indicator. It is clear that a larger magnitude of $p_1$ (or $p_2$) corresponds to a larger magnitude of correlation between $U$ and the low malaria prevalence indicator (or the low birth weight indicator). We now discuss how the proposed sensitivity analysis model helps to address concerns about the potential hidden bias from Perspectives 1–3 listed above:

- For Perspective 1: The proposed sensitivity analysis model covers Perspective 1 by considering a hypothetical unobserved covariate $U$ such that it is correlated with both the low malaria prevalence indicator (i.e. the indicator for units who have experienced sharp drops in malaria prevalence) (by prespecifying various $p_1$) and the low birth weight indicator (by prespecifying various $p_2$). With the unobserved covariate $U$, the unconfoundedness assumption may be violated as matching can only adjust for observed covariates but cannot directly adjust for unobserved covariates.
- For Perspective 2: The proposed sensitivity analysis model also covers Perspective 2 by including the unobserved covariate $U$ in the final outcome model. This is because by setting a non-zero $p_1$, the distributions of $U$ between high-low and high-high pairs of clusters will be imbalanced (i.e. will not follow a common trajectory). Meanwhile, by setting a non-zero $p_2$ (corresponds to a non-zero $\lambda$ in Model (3)), the imbalances of $U$ across the treated and controls will make the outcome trend of the high-low pairs of clusters (i.e. the treated group) in the absence of the treatment deviate from a parallel trend with that of the high-high pairs (i.e. the control group).
- For Perspective 3: When setting $p_1 \neq 0$, the hypothetical unobserved covariate $U$ in our sensitivity analysis model can also be regarded as some event of which the occurrence probability varies across the treated group and the control group and is not directly associated with observed covariates. Meanwhile, by setting some $p_2 \neq 0$, the contribution of that event to the low birth weight rate differs across the treated group and the control group as that event occurs more (or less) frequently in the treated group. Therefore, our sensitivity analyses also cover Perspective 3 of the potential hidden bias.

After setting up the sensitivity analysis model (3), the detailed sensitivity analysis procedure is as follows. For each pair of prespecified sensitivity parameters $(p_1, p_2)$ and for each imputed data set (500 in total) obtained from Step 3 (the multiple imputation stage), we generate the value of $U_{ij}$ for each individual $j$ in cluster $i$ according to Model (4) and calculate the corresponding point estimate and estimated standard error of the coefficient of the low malaria prevalence indicator under Model (3). Similarly to the primary analysis, for each pair of prespecified $(p_1, p_2)$, the corresponding estimated causal effect of reduced malaria burden on the low birth weight rate is the mean value of the 500 estimated coefficients on the low malaria prevalence indicator obtained from 500 runs of Model (3). The corresponding p-value and 95% CIs can also be obtained via applying Rubin's rules with treating the imputed $U$ as an usual regressor in Model (3). We conduct the above procedure for various $(p_1, p_2)$ and examine how the results differ from those in the primary analysis.

## Detailed results of the sensitivity analyses

When reporting the sensitivity analyses for the coefficient of the low malaria prevalence indicator under the sensitivity analysis model (3) with various prespecified values of the sensitivity parameters $(p_1, p_2)$, we divide the results into the following four cases:

- Case 1: $p_1 > 0, p_2 > 0$. That is, the hypothetical unobserved covariate $U$ is positively correlated with both the low malaria prevalence indicator (i.e., the indicator for units who have experienced sharp drops in malaria prevalence) and the low birth weight indicator (i.e., the outcome variable).
- Case 2: $p_1 > 0, p_2 < 0$. That is, the hypothetical unobserved covariate $U$ is positively correlated with the low malaria prevalence indicator while it is negatively correlated with the low birth weight indicator.

- Case 3: $p_1<0, p_2>0$. That is, the hypothetical unobserved covariate $U$ is negatively correlated with the low malaria prevalence indicator while it is positively correlated with the low birth weight indicator.
- Case 4: $p_1<0, p_2<0$. That is, the hypothetical unobserved covariate $U$ is negatively correlated with both the low malaria prevalence indicator and the low birth weight indicator.

We report the results of the sensitivity analyses in *Appendix 3—table 1*. Specifically, for each $(p_1, p_2)$, we report the point estimate, the 95% CI, and the p-value (under null effect) of the low malaria prevalence indicator under Model (3) in which the hypothetical unobserved covariate $U_{ij}$ is generated from Model (4) within each imputed data set.

We list the interpretations of the results in *Appendix 3—table 1* case by case:

- Cases 1 and 4: In these two cases, the magnitude of the estimated treatment effect obtained from the primary analysis assuming no observed covariates (1.48 percentage points reduction, listed in *Table 6* of the main text) is smaller than that obtained from the sensitivity analyses in which the unobserved covariate $U$ is taken into account. This implies that if the unobserved covariate more (or less) frequently appears in the treated group and predicts the outcome in the opposite (or same) direction as the treatment does, the primary analysis tends to underestimate the actual treatment effect. This pattern agrees with the previous literature on sensitivity analyses (*Rosenbaum and Silber, 2009*). However, as shown in *Appendix 3—table 1*, the magnitude of this potential estimation bias is estimated to be no greater than $|-1.83-(-1.48)| = 0.35$ percentage points as long as $p_1, p_2 \in (0, 10)$ percentage points or $p_1, p_2 \in [-10, 0)$ percentage points.
- Cases 2 and 3: In these two cases, the magnitude of the estimated treatment effect obtained from the primary analysis is smaller than that obtained from the sensitivity analyses with $U$ taken into account. This implies that if the unobserved covariate more (or less) frequently appears in the treated group and predicts the outcome in the same (or opposite) direction as the treatment does, the primary analysis tends to overestimate the actual treatment effect. This pattern also agrees with the previous literature on sensitivity analyses (*Rosenbaum and Silber, 2009*). However, as shown in *Appendix 3—table 1*, the magnitude of this potential estimation bias is estimated to be no greater than $|-1.13-(-1.48)| = 0.35$ percentage points as long as $|p_1| \le 10$ percentage points and $|p_2| \le 10$ percentage points.

**Appendix 3—table 1.** The results of the sensitivity analyses for the coefficient of the low malaria prevalence indicator under various sensitivity parameters $(p_1, p_2)$ divided into the four cases: Case 1: $p_1>0, p_2>0$; Case 2: $p_1>0, p_2<0$; Case 3: $p_1<0, p_2>0$; Case 4: $p_1<0, p_2<0$.

The unit of estimates and CIs is a percentage point.

| Case 1 | $p_2 = 5.0$ | | | $p_2 = 10.0$ | | |
|---|---|---|---|---|---|---|
| | Estimate | 95% CI | p-value | Estimate | 95% CI | p-value |
| $p_1 = 2.5$ | −1.52 | [−3.74, 0.70] | 0.179 | −1.56 | [−3.77, 0.66] | 0.168 |
| $p_1 = 5.0$ | −1.57 | [−3.79, 0.66] | 0.167 | −1.65 | [−3.86, 0.57] | 0.145 |
| $p_1 = 7.5$ | −1.61 | [−3.83, 0.61] | 0.156 | −1.73 | [−3.95, 0.48] | 0.125 |
| $p_1 = 10.0$ | −1.65 | [−3.88, 0.57] | 0.145 | −1.82 | [−4.04, 0.40] | 0.107 |
| Case 2 | $p_2 = -5.0$ | | | $p_2 = -10.0$ | | |
| | Estimate | 95% CI | p-value | Estimate | 95% CI | p-value |
| $p_1 = 2.5$ | −1.44 | [−3.66, 0.78] | 0.204 | −1.39 | [−3.62, 0.83] | 0.219 |
| $p_1 = 5.0$ | −1.40 | [−3.62, 0.83] | 0.218 | −1.31 | [−3.53, 0.92] | 0.249 |
| $p_1 = 7.5$ | −1.35 | [−3.58, 0.87] | 0.234 | −1.22 | [−3.44, 1.00] | 0.282 |
| $p_1 = 10.0$ | −1.31 | [−3.53, 0.92] | 0.250 | −1.13 | [−3.36, 1.09] | 0.318 |
| Case 3 | $p_2 = 5.0$ | | | $p_2 = 10.0$ | | |
| | Estimate | 95% CI | p-value | Estimate | 95% CI | p-value |
| $p_1 = -2.5$ | −1.44 | [−3.66, 0.78] | 0.204 | −1.39 | [−3.61, 0.83] | 0.219 |
| $p_1 = -5.0$ | −1.39 | [−3.61, 0.83] | 0.219 | −1.30 | [−3.52, 0.91] | 0.249 |

*Continued on next page*

| | Estimate | 95% CI | p-value | Estimate | 95% CI | p-value |
|---|---|---|---|---|---|---|
| $p_1 = -7.5$ | −1.35 | [−3.57, 0.87] | 0.234 | −1.22 | [−3.43, 1.00] | 0.282 |
| $p_1 = -10.0$ | −1.31 | [−3.53, 0.92] | 0.250 | −1.13 | [−3.35, 1.09] | 0.319 |
| | $p_2 = -5.0$ | | | $p_2 = -10.0$ | | |
| **Case 4** | **Estimate** | **95% CI** | **p-value** | **Estimate** | **95% CI** | **p-value** |
| $p_1 = -2.5$ | −1.52 | [−3.75, 0.70] | 0.179 | −1.56 | [−3.79, 0.66] | 0.168 |
| $p_1 = -5.0$ | −1.57 | [−3.79, 0.66] | 0.167 | −1.65 | [−3.87, 0.57] | 0.146 |
| $p_1 = -7.5$ | −1.61 | [−3.84, 0.61] | 0.156 | −1.74 | [−3.96, 0.49] | 0.126 |
| $p_1 = -10.0$ | −1.66 | [−3.88, 0.57] | 0.145 | −1.83 | [−4.05, 0.40] | 0.108 |

