## [Decision Letter]

**Acceptance summary:**

This paper provides relevant findings of the effects of changing malaria burden on low birth weight using a novel design-based causal inference approach (i.e., a two-step matching procedure as nonparametric data preprocessing in a difference-in-differences design). There are a lot of things to like about this paper, such as its creative design, its extensive data collection effort, and its important results for the policy and health literature. The research topic is well-motivated and of importance to the malaria research community. The statistical methods will be applicable in other contexts.

**Decision letter after peer review:**

Thank you for submitting your article "Relationship between changing malaria burden and low birth weight in sub-Saharan Africa" for consideration by *eLife*. Your article has been reviewed by 2 peer reviewers, and the evaluation has been overseen by a Reviewing Editor and a Senior Editor. The following individual involved in review of your submission has agreed to reveal their identity: Jennifer Flegg (Reviewer #1).

Summary:

This paper provides relevant findings of the effects of changing malaria burden on low birth weight using a novel design-based causal inference approach (i.e., a two-step matching procedure as nonparametric data preprocessing in a difference-in-differences design). There are a lot of things to like about this paper, such as its creative design, its extensive data collection effort, and its important results for the policy and health literature. The research topic is well-motivated and of importance to the malaria research community. The statistical methods will be applicable in other contexts.

Essential Revisions:

1. To answer the research question, the authors make use of the fact that "while the overall prevalence of malaria has declined in sub-Saharan Africa, the decline has been uneven, with some malaria-endemic areas experiencing sharp drops and others experiencing little change" (page 2). However, they never discuss if that heterogeneity can be explained by hidden variables that could also predict the outcomes. In other words, an unobserved factor u could be explaining both the uneven reduction of malaria in certain areas and birth weight. I understand that it is impossible to provide a final answer to this concern, but I would expect some discussion about it. Probably a sensitivity analysis or the amplification of a sensitivity analysis could help to shed some light on this issue.

2. The authors combine two strategies to answer the research question: difference-in-differences and matching. The authors justify combining both strategies by claiming that matching can make the parallel trend assumption more likely to hold. Even though this point makes sense, it would require a better explanation. The main assumption behind a DID is the outcome in treated and control groups would follow parallel trends in the absence of the treatment. As a result, both observed and unobserved characteristics should also follow a common trajectory across both the control and treated groups before the intervention. Matching can help to address imbalances in terms of observed covariates, however, it will require authors to make assumptions about unobservables that are not discussed in the paper. As in my previous comment, a sensitivity analysis might help to address this concern or at least a discussion to better understand under what conditions matching can credibility enhance a difference-in-differences design.

3. Locations are different across time within the same country because the DHS sample different places when constructing representative samples. This is not a problem per se when implementing a DID. However, it might become problematic if the sampling variability is generating imbalances in only one group and therefore those imbalances are not following a common trajectory, which might produce biased results when implementing the DID. In other words, it is not an issue if sampling variability is changing both the control and treated groups in the same direction, but it is going to be an issue if that is only happening for one of the groups or for both groups but in opposite directions. Matching can provide a direct solution to this problem since now we can attribute the changes in outcomes to the intervention and not to a different group composition. I would recommend authors to discuss how matching can help to improve difference-in-differences design when the units of analysis are not the same across time.

4. DID will be biased if there is an event that is occurring at the same time as the intervention, and therefore that is only affecting the treated group after the treatment (i.e., selective maturation). For example, places experiencing a malaria prevalence decline might also experience other positive health outcomes that can contribute to explaining birth weight. I would encourage authors to discuss this possibility.

5. There has been updated estimates of pf parasite rates that include estimates up to 2017 which I think should be used.

6. The choice of "early" and "late" years was a little arbitrary – is there any justification that can be provided or a sensitivity on these definitions?

7. It wasn't clear how the sociodemographic covariates were chosen – is this list (on page 7) an exhaustive list? If not, how were these covariates chosen?

8. In how the model in Equation (1) is presented, that the mixed effects model was fitted in a frequentist setting? Eg there is a reference later to confidence intervals (not credible intervals). I find the change of using a Bayesian approach in the earlier steps of the methodology to using a Frequentist approach quite strange and would suggest that a consistent approach is used throughout.

9. One of the main findings is about the reduction in the rate of low birth weight, but this is not (at the 95% confidence level) significant based on the modelling. While this is mentioned in the Discussion of the paper, I think this should be more up-front in the abstract/results.

10. Can you share on github or similar?

Reviewer #1:

This paper presents a statistical approach to quantify the relationship between changing pf malaria and the rate of a low-birth weight babies in Africa. A major strength of the work is that the methodology brings together multiple data sources and statistical methods into the one analysis. The authors have achieved their aims and the results support their conclusions. The research topic is well-motivated and of importance to the malaria research community. The statistical methods will be applicable in other contexts.

1. There has been updated estimates of pf parasite rates that include estimates up to 2017 which I think should be used.

2. I thought the choice of "early" and "late" years was a little arbitrary – is there any justification that can be provided or a sensitivity on these definitions?

3. It wasn't clear how the sociodemographic covariates were chosen – is this list (on page 7) an exhaustive list? If not, how were these covariates chosen?

4. It seems to me, in how the model in Equation (1) is presented, that the mixed effects model was fitted in a frequentist setting? Eg there is a reference later to confidence intervals (not credible intervals). I find the change of using a Bayesian approach in the earlier steps of the methodology to using a Frequentist approach quite strange and would suggest that a consistent approach is used throughout.

5. One of the main findings is about the reduction in the rate of low birth weight, but this is not (at the 95% confidence level) significant based on the modelling. While this is mentioned in the Discussion of the paper, I think this should be more up-front in the abstract/results.

Reviewer #2:

The authors study the effects of a malaria prevalence decline on low birth rates. To achieve this goal, they use data from 19 countries in sub-Saharan Africa. This manuscript's main strengths are the data collection efforts (merging annual malaria prevalence, demographic and health surveys, and geographical information) and the use of a novel methodological approach (combining recent developments in optimal matching with a difference-in-differences design). Some weaknesses or aspects that could be improved are the lack of discussion about possible biases such as the presence of unobserved factors that could explain the uneven decline in malaria, the role of sampling variability when using survey data to construct a difference-in-differences design, and the role of selective maturation. Regardless of these previous points, the paper makes clear contributions to the applied causal inference literature by illustrating how it is possible to enhance a traditional difference-in-differences design, and to the health and policy literature by showing the effects of malaria on a crucial development outcome.

I have four main comments/suggestions for the authors:

1. To answer the research question, the authors make use of the fact that "while the overall prevalence of malaria has declined in sub-Saharan Africa, the decline has been uneven, with some malaria-endemic areas experiencing sharp drops and others experiencing little change" (page 2). However, they never discuss if that heterogeneity can be explained by hidden variables that could also predict the outcomes. In other words, an unobserved factor u could be explaining both the uneven reduction of malaria in certain areas and birth weight. I understand that it is impossible to provide a final answer to this concern, but I would expect some discussion about it. Probably a sensitivity analysis or the amplification of a sensitivity analysis could help to shed some light on this issue.

2. The authors combine two strategies to answer the research question: difference-in-differences and matching. The authors justify combining both strategies by claiming that matching can make the parallel trend assumption more likely to hold. Even though this point makes sense, it would require a better explanation. The main assumption behind a DID is the outcome in treated and control groups would follow parallel trends in the absence of the treatment. As a result, both observed and unobserved characteristics should also follow a common trajectory across both the control and treated groups before the intervention. Matching can help to address imbalances in terms of observed covariates, however, it will require authors to make assumptions about unobservables that are not discussed in the paper. As in my previous comment, a sensitivity analysis might help to address this concern or at least a discussion to better understand under what conditions matching can credibility enhance a difference-in-differences design.

3. Locations are different across time within the same country because the DHS sample different places when constructing representative samples. This is not a problem per se when implementing a DID. However, it might become problematic if the sampling variability is generating imbalances in only one group and therefore those imbalances are not following a common trajectory, which might produce biased results when implementing the DID. In other words, it is not an issue if sampling variability is changing both the control and treated groups in the same direction, but it is going to be an issue if that is only happening for one of the groups or for both groups but in opposite directions. Matching can provide a direct solution to this problem since now we can attribute the changes in outcomes to the intervention and not to a different group composition. I would recommend authors to discuss how matching can help to improve difference-in-differences design when the units of analysis are not the same across time.

4. DID will be biased if there is an event that is occurring at the same time as the intervention, and therefore that is only affecting the treated group after the treatment (i.e., selective maturation). For example, places experiencing a malaria prevalence decline might also experience other positive health outcomes that can contribute to explaining birth weight. I would encourage authors to discuss this possibility.

---

## [Author Response]

Reviewer #1:This paper presents a statistical approach to quantify the relationship between changing pf malaria and the rate of a low-birth weight babies in Africa. A major strength of the work is that the methodology brings together multiple data sources and statistical methods into the one analysis. The authors have achieved their aims and the results support their conclusions. The research topic is well-motivated and of importance to the malaria research community. The statistical methods will be applicable in other contexts.1. There has been updated estimates of pf parasite rates that include estimates up to 2017 which I think should be used.

Thank you for checking this issue with us and for these helpful instructions and suggestions in the decision letter related to this comment. Our response to this comment and the related comments from the decision letter consists of three parts: 1) clarifications on why we chose the years 2000–2015 as the study period of our study at the design stage and why we stick with this choice throughout the outcome analysis in the current study; 2) what adjustments corresponding to these comments have been made in the revised manuscript; 3) the confirmation that we plan to conduct a follow-up study in which those published or upcoming post-2015 Malaria Atlas Project (MAP) data will also be considered and the resulting paper or technical report will be linked to the final published version of the current paper.

First, we here clarify why we chose the years 2000–2015 as the study period at the study design stage and then stick with this choice throughout the outcome analysis of this study. We chose the year 2000 as the starting point of the study period because it is both the earliest year in which the estimated annual *Pf* PR_2−10_ is published by the Malaria Atlas Project (MAP) (MAP, 2020) and a turning point year in the multilateral commitment to malaria control in sub-Saharan Africa (Bhatt et al., 2015). We chose the year 2015 as the ending point of the study period is mainly based on the fact that at the design stage of our study (the year 2017), the year 2015 was the latest year in which the estimated annual *Pf* PR_2−10_ produced by MAP was available to us. As you have pointed out, MAP published some post-2015 estimated annual *Pf* PR_2−10_ data since then. However, including these post-2015 data in the midst of our outcome statistical inference procedure will change the study sample determined at the design stage and therefore may influence the transparency and validity of our statistical inference (Rubin, 2007). Following Rubin (2007)’s advice to design observational studies before seeing and analyzing the outcome data, throughout the whole outcome analysis of the current study, we stick with the study sample selection procedure determined at the design stage and do not change the prespecified study period (the years 2000–2015) to ensure the transparency of the statistical inference of the current study.

Second, we list what corresponding adjustments have been made in the revised manuscript from the following three aspects:

1. To make the reasons why we chose the years 2000–2015 as the study period at the design stage and then stick with this choice throughout the whole outcome analysis of the current study more transparent for readers, we have added the following paragraph at the beginning of the “Data selection procedure” section (on page 4 of the revised manuscript):

“In this article, we set the study period to be the years 2000–2015, and correspondingly, all the results and conclusions obtained in this article are limited to the years 2000–2015. We set the year 2000 as the starting point of the study period for two reasons. First, the year 2000 is the earliest year in which the estimated annual *Pf* PR_2−10_ is published by MAP (MAP, 2020). Second, according to Bhatt et al. (2015), “the year 2000 marked a turning point in multilateral commitment to malaria control in sub-Saharan Africa, catalysed by the Roll Back Malaria initiative and the wider development agenda around the United Nations Millennium Development Goals.” We set the year 2015 as the ending point based on two considerations. First, when we designed our study in the year 2017, the year 2015 was the latest year in which the estimated annual *Pf* PR_2−10_ was available to us. We became aware after starting our outcome analysis that MAP has published some post-2015 estimated annual *Pf* PR_2−10_ data since then, but, following Rubin (2007)’s advice to design observational studies before seeing and analyzing the outcome data, we felt it was best to stick with the design of our original study for this report and consider the additional data in a later report. Second, the year 2015 was set as a target year by a series of international goals on malaria control. For example, the United Nations Millennium Development Goals set a goal to “halt by 2015 and begin to reverse the incidence of malaria” and ``the more ambitious target defined later by the World Health Organization (WHO) of reducing case incidence by 75% relative to 2000 levels." (WHO, 2008). It is worth emphasizing that although we set the years 2000–2015 as the study period and did not investigate any post-2015 MAP data because of the above considerations, those published or upcoming post-2015 MAP data should be considered or leveraged for future related research or follow-up studies.”

2. Following the suggestion corresponding to your Comment 1 in the decision letter, to limit our results and conclusions drawn from this article to the years 2000–2015 and make this point more transparent for readers, we have made the following adjustments in the revised manuscript:

i. In the “Abstract” section (on page 1 of the revised manuscript), we have added the words “during the years 2000–2015” in the sentence “We conducted an observational study of the effect of changing malaria burden on low birth weight using data from 18,112 births in 19 countries in sub-Saharan African countries during the years 2000–2015.”

ii. We have added the following sentence at the beginning of the “Data selection procedure” section (on page 4 of the revised manuscript): “In this article, we set the study period to be the years 2000–2015, and correspondingly, all the results and conclusions obtained in this article are limited to the years 2000–2015.”

iii. We have added the following sentence at the beginning of the “Results” section (on page 14 of the revised manuscript): “In this section, we report and interpret the results of matching, primary analysis, secondary analyses, and sensitivity analyses relating changes in malaria burden to changes in the birth weight rate between 2000–2015 in sub-Saharan Africa.”

iv. At the beginning of the “Discussion” section (on page 18 of the revised manuscript), we have added the words “during the years 2000– 2015” in the sentence “We have developed a pair-of-pairs matching approach to conduct a difference-in-differences study to examine the causal effect of a reduction in malaria prevalence on the low birth weight rate in sub-Saharan Africa during the years 2000–2015.”

3. To emphasize that although we did not include any post-2015 MAP data in the current study for the transparency of our statistical inference, these post-2015 MAP data are useful and should be considered in future research, we have included the following sentence in the “Data selection procedure” section (on page 4 of the revised manuscript): “It is worth emphasizing that although we set the years 2000–2015 as the study period and did not investigate any post-2015 MAP data because of the above considerations, those published or upcoming post-2015 MAP data should be considered or leveraged for future related research or follow-up studies.”

Third, we here confirm that, following *eLife*’s requirement, we will conduct a followup study in the future to consider both the 2000–2015 MAP data and any published or upcoming post-2015 MAP data and report how including the additional data would affect the relevant conclusions. We are considering two types of follow-up study determined by our further investigation on the study design and the data selection procedure:

1. Type-I follow-up study: We follow a similar study design as that in the current paper except that this time we consider both the 2000–2015 MAP data and any published or upcoming post-2015 MAP data. This type of follow-up study would be more appropriate to be developed into a short technical report posted on a preprint server.

2. Type-II follow-up study: We both consider the relevant additional new data (including those post-2015 MAP data) and elaborate on the current study design and statistical approach. That is, this type of follow-up study aims to contribute to both the empirical research problem and the statistical methodology. This type of follow-up study would be more appropriate to be developed into a formal paper which would be both posted on a preprint server and sent out for peer review.

2. I thought the choice of "early" and "late" years was a little arbitrary – is there any justification that can be provided or a sensitivity on these definitions?

Thank you for checking this issue with us. To clarify this issue and provide some justifications on the robustness of the study to our way of defining the “early years” and the “late years,” we have made the following adjustments to the manuscript:

We have added the following sentences to the “Data selection procedure” section (on page 4 of the revised manuscript) to clarify how we define the “early years” and the “late years”: “After selecting 2000–2015 as our study period, we take the middle point years 2007 and 2008 as the cut-off and define the years 2000–2007 as the “early years” and the years 2008–2015 as the “late years.””

To assess the sensitivity of the study to our way of defining the “early years” and the “late years,” we added the following paragraph to interpret Table 1 (on pages 4 and 5 of the revised manuscript): “From Table 1, we can see that among the 19 countries, only two countries (Congo Democratic Republic and Zambia) happen to take the margin year 2007 as the early year and no countries take the margin year 2008 as the late year. This implies that our study is relatively insensitive to our way of defining the early years (2000–2007) and the late years (2008–2015) as most of the selected early years and late years in Table 1 do not fall near the margin years 2007 and 2008.”

3. It wasn't clear how the sociodemographic covariates were chosen – is this list (on page 7) an exhaustive list? If not, how were these covariates chosen?

Thank you for checking this issue with us. To clarify this issue, we have added the following paragraph below the list of the six chosen sociodemographic covariates:

“The above six sociodemographic covariates were chosen by looking over the variables in the Demographic and Health Surveys (DHS) and choosing those which we thought met the following criteria: 1) The above six covariates are potentially strongly correlated with both the risk of malaria (Baragatti et al., 2009; Krefis et al., 2010; Ayele et al., 2013; Roberts and Matthews, 2016; Sulyok et al., 2017) and birth outcomes (Sahn and Stifel, 2003; Gemperli et al., 2004; Chen et al., 2009; Grace et al., 2015; Padhi et al., 2015), and therefore may be important confounding variables that need to be adjusted for via statistical matching (Rosenbaum and Silber, 2009; Rosenbaum, 2010; Stuart, 2010). 2) The records of the above six covariates are mostly available for all the countries and the survey years in our study samples. Specifically, for the above six covariates, the percentages of missing data (missingness can arise either because the question was not asked or the individual was asked the question but did not respond) among the total individual records from IPUMS-DHS among the 6,812 pairs of clusters remaining after Step 1 are all less than 0.3%.”

4. It seems to me, in how the model in Equation (1) is presented, that the mixed effects model was fitted in a frequentist setting? Eg there is a reference later to confidence intervals (not credible intervals). I find the change of using a Bayesian approach in the earlier steps of the methodology to using a Frequentist approach quite strange and would suggest that a consistent approach is used throughout.

Thank you for checking this issue with us. To clarify this issue, we have added the following paragraph at the end of the “Statistical analysis” section (on page 12 of the revised manuscript):

“It is worth clarifying that although we take a Bayesian approach when imputing (i.e., predicting) the missing low birth weight indicators in Step 3 (i.e., imputation model) and then take a frequentist approach when conducting the 500 separate outcome analyses with the 500 imputed data sets in Step 4 (i.e., substantive model), these two different statistical perspectives (i.e., Bayesian and frequentist) do not conflict with each other when we apply Rubin’s rules to combine these 500 separate outcome analyses as the single estimator and inference reported in Table 6. This is because the frequentist validity of applying Rubin’s rules to combine separate outcome analyses with multiple imputed data sets only explicitly depends on the asymptotic normal approximation assumption for each coefficient estimator in Model (1) (see Appendix 2 for more details), and does not directly depend on how the multiple imputed data sets are generated (e.g., either using a Bayesian imputation model as in Step 3 or using a frequentist imputation model instead). Using a Bayesian imputation model followed by a frequentist substantive model is one of the most common strategies when applying Rubin’s rules to conduct statistical inference with multiple imputation; see Rubin (1996), Chapter 3 of Rubin (1987), and Chapter 2 of Carpenter and Kenward (2012). For representative works on justifying the advantages of using a Bayesian imputation model in multiple-imputation inferences, see Meng (1994) and Chapter 2 of Carpenter and Kenward (2012).”

5. One of the main findings is about the reduction in the rate of low birth weight, but this is not (at the 95% confidence level) significant based on the modelling. While this is mentioned in the Discussion of the paper, I think this should be more up-front in the abstract/results.

Thank you for this helpful suggestion. We have made the following adjustments correspondingly in the revised manuscript:

We have added the following sentence to the abstract (on page 1 of the revised manuscript): “Although the confidence intervals cannot rule out the possibility of no effect at the 95% confidence level, the concurrence between our primary analysis, secondary analyses, and sensitivity analyses, and the magnitude of the effect size, contribute to the weight of the evidence suggesting that declining malaria burden can potentially substantially reduce the low birth weight rate at the community level in sub-Saharan Africa, particularly among firstborns.”

We have added the following paragraph at the end of the “Results” section (on page 18 of the revised manuscript): “To conclude, although the confidence intervals of the coefficient of the low malaria prevalence indicator on the low birth weight rate presented in the “Results” section cannot exclude a possibility of no effect at level 95% based on our proposed study sample selection procedure and statistical approach, the results and the corresponding interpretations of the primary analysis, the secondary analyses, and the sensitivity analyses have contributed to the weight of the evidence that reduced malaria burden has an important influence on the low birth weight rate in sub-Saharan Africa at the community level.”

Reviewer #2:The authors study the effects of a malaria prevalence decline on low birth rates. To achieve this goal, they use data from 19 countries in sub-Saharan Africa. This manuscript's main strengths are the data collection efforts (merging annual malaria prevalence, demographic and health surveys, and geographical information) and the use of a novel methodological approach (combining recent developments in optimal matching with a difference-in-differences design). Some weaknesses or aspects that could be improved are the lack of discussion about possible biases such as the presence of unobserved factors that could explain the uneven decline in malaria, the role of sampling variability when using survey data to construct a difference-in-differences design, and the role of selective maturation. Regardless of these previous points, the paper makes clear contributions to the applied causal inference literature by illustrating how it is possible to enhance a traditional difference-in-differences design, and to the health and policy literature by showing the effects of malaria on a crucial development outcome.I have four main comments/suggestions for the authors:1. To answer the research question, the authors make use of the fact that "while the overall prevalence of malaria has declined in sub-Saharan Africa, the decline has been uneven, with some malaria-endemic areas experiencing sharp drops and others experiencing little change" (page 2). However, they never discuss if that heterogeneity can be explained by hidden variables that could also predict the outcomes. In other words, an unobserved factor u could be explaining both the uneven reduction of malaria in certain areas and birth weight. I understand that it is impossible to provide a final answer to this concern, but I would expect some discussion about it. Probably a sensitivity analysis or the amplification of a sensitivity analysis could help to shed some light on this issue.

Thank you for this helpful suggestion. We have added two new sections “Sensitivity analyses” (on pages 13 and 14 of the revised manuscript) and “Results of the sensitivity analyses” (on page 18 of the revised manuscript) to the main text and two new sections “Design of the sensitivity analyses” (on pages 32–34 of the revised manuscript) and “Detailed results of the sensitivity analyses” (on pages 34 and 35 of the revised manuscript) to Appendix 3 to address the hidden bias issues raised in your Comments 1, 2, and 4. Specifically, we realize that these three comments correspond to three closely connected perspectives on potential hidden bias in the previous literature that cannot be directly removed by neither matching nor a difference-in-differences design (see Perspectives 1–3 in the “Sensitivity analyses” section of the main text and the “Design of the sensitivity analyses” section of Appendix 3), and therefore should be addressed by sensitivity analyses. Among Perspectives 1–3 in the newly added “Sensitivity analyses” section and the “Design of the sensitivity analyses” section, Perspective 1 corresponds to your Comment 1 and is briefly summarized in the “Sensitivity analyses” section written as follows (on page 13 of the revised manuscript): “Perspective 1: The potential violation of the unconfoundedness assumption (Rosenbaum and Rubin, 1983; Heckman and Robb, 1985; Heckman et al., 1997).” We then give a detailed description of Perspective 1 in the “Design of the sensitivity analyses” section of Appendix 3 (on page 32 of the revised manuscript) written as follows:

“Perspective 1: The potential violation of the unconfoundedness assumption (Rosenbaum and Rubin, 1983; Heckman and Robb, 1985). Roughly speaking, the unconfoundedness assumption states that, after adjusting for observed covariates (measured confounders), there are no differential trends over time of any characteristics, other than the intervention itself, between the treated group and the control group, that may be correlated with their outcomes. This assumption may be violated if there is selection bias on unobserved covariates across time (Heckman and Robb, 1985; Heckman et al., 1997) such that there are differences in these observed covariates of the treated group and the control group which impact their trends in the outcome (Ashenfelter and Card, 1984; Doyle et al., 2018). For example, in our study, the unconfoundedness assumption can be violated if the sharp drops in malaria prevalence experienced by some areas could be explained by the changes of some unobserved characteristics over time that could also predict the low birth weight rate.”

To assess the robustness of the results of our primary analysis to potential hidden bias which can be viewed from various alternative perspectives (including Perspectives 1–3 described in the “Sensitivity analyses” section and the “Design of the sensitivity analyses” section), we have proposed a sensitivity analysis model in which we include a hypothetical unobserved covariate (unmeasured confounding variable or event) *U* that is correlated with both the low malaria prevalence indicator and the low birth weight indicator (see pages 14 and 33 of the revised manuscript). In Appendix 3 (on page 33 of the revised manuscript), we have discussed how our sensitivity analysis model covers Perspectives 1–3 when assessing the potential hidden bias. We post the corresponding interpretation of our sensitivity analysis model for addressing Perspective 1:

“For Perspective 1: The proposed sensitivity analysis model covers Perspective 1 by considering a hypothetical unobserved covariate *U* such that it is correlated with both the low malaria prevalence indicator (i.e., the indicator for units who have experienced sharp drops in malaria prevalence) (by prespecifying various *p*_1_) and the low birth weight indicator (by prespecifying various *p*_2_). With the unobserved covariate *U*, the unconfoundedness assumption may be violated as matching can only adjust for observed covariates but cannot directly adjust for unobserved covariates.”

Results of the sensitivity analyses are briefly summarized in the newly added “Results of the sensitivity analyses” section (on page 18 of the revised manuscript) and are reported in detail in Appendix 3 (on pages 34 and 35 of the revised manuscript). Specifically, in Table 1 of Appendix 3, we report the point estimates, the 95% confidence intervals, and the *p*-values of the coefficient of the low malaria prevalence indicator (i.e., the treatment effect) under the proposed sensitivity analysis model with various magnitudes of unmeasured confounding. In conclusion, our sensitivity analyses suggest that the magnitude of the estimated treatment effect is still evident even if the magnitude of unmeasured confounding was nontrivial. Please see pages 18, 34, and 35 of the revised manuscript for details on the results and interpretations of the sensitivity analyses.

2. The authors combine two strategies to answer the research question: difference-in-differences and matching. The authors justify combining both strategies by claiming that matching can make the parallel trend assumption more likely to hold. Even though this point makes sense, it would require a better explanation. The main assumption behind a DID is the outcome in treated and control groups would follow parallel trends in the absence of the treatment. As a result, both observed and unobserved characteristics should also follow a common trajectory across both the control and treated groups before the intervention. Matching can help to address imbalances in terms of observed covariates, however, it will require authors to make assumptions about unobservables that are not discussed in the paper. As in my previous comment, a sensitivity analysis might help to address this concern or at least a discussion to better understand under what conditions matching can credibility enhance a difference-in-differences design.

Thank you for this helpful suggestion. We have added two new sections “Sensitivity analyses” (on pages 13 and 14 of the revised manuscript) and “Results of the sensitivity analyses” (on page 18 of the revised manuscript) to the main text and two new sections “Design of the sensitivity analyses” (on pages 32–34 of the revised manuscript) and “Detailed results of the sensitivity analyses” (on pages 34 and 35 of the revised manuscript) to Appendix 3 to address the issues raised in your Comments 1, 2, and 4. Specifically, among Perspectives 1–3 on the potential hidden bias in the newly added “Sensitivity analyses” and “Design of sensitivity analyses” sections, Perspective 2 corresponds to your Comment 2 and is briefly summarized as follows (on page 13 of the revised manuscript): “Perspective 2: The potential violation of the parallel trend assumption in a difference-in-differences study (Card and Krueger, 2000; Angrist and Pischke, 2008; Hasegawa et al., 2019; Basu and Small, 2020).” We then give a detailed description of Perspective 2 in the “Design of the sensitivity analyses” section of Appendix 3 (on page 32 of the revised manuscript) written as follows:

“Perspective 2: The potential violation of the parallel trend assumption in a differencein-differences study (Card and Krueger, 2000; Angrist and Pischke, 2008; Hasegawa et al., 2019; Basu and Small, 2020). Recall that the parallel trend assumption behind a difference-in-differences study states that, in the absence of the treatment (i.e., intervention), after adjusting for relevant covariates, the outcome trajectory of the treated group would follow a parallel trend with that of the control group. Therefore, to make the parallel trend assumption more likely to hold, ideally each observed or unobserved covariate should be well balanced (i.e., follow a common trajectory) between the treated group and the control group, before and after the intervention. Matching can balance observed covariates by ensuring each covariate follows a common trajectory in the treated and control groups. However, matching cannot directly adjust for unobserved covariates and their trajectories among the treated and control groups may differ and correspondingly the parallel trend assumption may not hold.”

We have designed a sensitivity analysis model (on pages 14 and 33 of the revised manuscript) in which we include a hypothetical unobserved covariate *U*. On page 33 of the revised manuscript, we have discussed in detail how our sensitivity analysis model covers Perspective 2 which corresponds to your Comment 2:

“For Perspective 2: The proposed sensitivity analysis model also covers Perspective 2 by including the unobserved covariate *U* in the final outcome model. This is because by setting a non-zero *p*_1_, the distributions of *U* between high-low and high-high pairs of clusters will be imbalanced (i.e., will not follow a common trajectory). Meanwhile, by setting a non-zero *p*_2_ (corresponds to a non-zero *λ* in Model (3)), the imbalances of *U* across the treated and controls will make the outcome trend of the high-low pairs of clusters (i.e., the treated group) in the absence of the treatment deviate from a parallel trend with that of the high-high pairs (i.e., the control group).”

We have reported the detailed results of the sensitivity analyses in Table 1 of Appendix 3 (on page 35 of the revised manuscript). Our sensitivity analyses in general support the conclusion from the primary analysis that the estimated magnitude of the treatment effect is evident, even under a nontrivial magnitude of unmeasured confounding. Please see pages 18, 34, and 35 of the revised manuscript for details.

3. Locations are different across time within the same country because the DHS sample different places when constructing representative samples. This is not a problem per se when implementing a DID. However, it might become problematic if the sampling variability is generating imbalances in only one group and therefore those imbalances are not following a common trajectory, which might produce biased results when implementing the DID. In other words, it is not an issue if sampling variability is changing both the control and treated groups in the same direction, but it is going to be an issue if that is only happening for one of the groups or for both groups but in opposite directions. Matching can provide a direct solution to this problem since now we can attribute the changes in outcomes to the intervention and not to a different group composition. I would recommend authors to discuss how matching can help to improve difference-in-differences design when the units of analysis are not the same across time.

Thank you for this helpful suggestion. When discussing how using matching as a data preprocessing step in a difference-in-differences design goes beyond directly applying a difference-in-differences approach in the “Motivation and overview of our approach: difference-in-differences via pair-of-pairs” section (on pages 6 and 7 of the revised manuscript), we have added the following paragraph to discuss how matching helps to address the survey location sampling variability issue when using survey data to conduct a difference-in-differences design raised in your Comment 3 (on pages 6 and 7 of the revised manuscript):

“Another perspective on how our second-step matching helps to improve a differencein-differences study is through the survey location sampling variability (Fakhouri et al., 2020). Recall that when constructing representative samples, the DHS are sampled at different locations (i.e., clusters) across time (ICF, 2019; Elizabeth Heger Boyle and Sobek, 2019). Therefore, if we simply implemented a difference-in-differences approach over all the high-low and high-high pairs of survey clusters and did not use matching to adjust for observed covariates, this survey location sampling variability may generate imbalances (i.e., different trajectories) of observed covariates across the treated and control groups, and therefore may bias the difference-in-differences estimator (Heckman et al., 1997). Imbalances of observed covariates caused by the survey location sampling variability may occur in the following three cases: 1) The survey location sampling variability is affecting the treated and control groups in the opposite direction. Specifically, there is some observed covariate for which the difference between the high-low pairs of sampled clusters tends to be larger (or smaller) than the country’s overall difference between the high malaria prevalence regions in the early years and the low malaria prevalence regions in the late years and conversely, the difference in that observed covariate between the high-high pairs of sampled clusters tends to be smaller (or larger) than the country’s overall difference between the high malaria prevalence regions in the early years and the high malaria prevalence regions in the late years. 2) The survey location sampling variability is affecting the treated and control groups in the same direction but to different extents. 3) The survey location sampling variability only happened in the treated or control group. Specifically, there is some observed covariate for which the difference between the high-low (or high-high) pairs of sampled clusters tends to differ from the country’s overall difference between the high malaria prevalence regions in the early years and the low (or high) malaria prevalence regions in the late years, but this is not the case for the high-high (or high-low) pairs of sampled clusters. Using matching as a nonparametric data preprocessing step in a difference-in-differences study can remove this type of bias because the observed covariates trajectories are forced to be common among the matched treated and control groups (St.Clair and Cook, 2015; Basu and Small, 2020).”

4. DID will be biased if there is an event that is occurring at the same time as the intervention, and therefore that is only affecting the treated group after the treatment (i.e., selective maturation). For example, places experiencing a malaria prevalence decline might also experience other positive health outcomes that can contribute to explaining birth weight. I would encourage authors to discuss this possibility.

Thank you for this helpful suggestion. As also mentioned above, we have added two new sections “Sensitivity analyses” (on pages 13 and 14 of the revised manuscript) and “Results of the sensitivity analyses” (on page 18 of the revised manuscript) to the main text and two new sections “Design of the sensitivity analyses” (on pages 32–34 of the revised manuscript) and “Detailed results of the sensitivity analyses” (on pages 34 and 35 of the revised manuscript) to Appendix 3. Specifically, among Perspectives 1–3 included in the newly added “Sensitivity analyses” and “Design of the sensitivity analyses” sections, Perspective 3 corresponds to your Comment 4 and is summarized as follows (on page 13 of the revised manuscript): “Perspective 3: The difference-in-differences estimator may be biased if there is an event that is more (or less) likely to occur as the intervention happens and the occurrence probability of this event cannot be fully captured by observed covariates (Shadish, 2010; West and Thoemmes, 2010).” We then give a detailed description of Perspective 3 in Appendix 3 (on page 32 of the revised manuscript):

“Perspective 3: A difference-in-differences study may be biased if there is an event that is more (or less) likely to occur as the treatment (i.e., intervention) happens in the treated group, but, unlike the case discussed in Section “Motivation and overview of our approach: difference-in-differences via pair-of-pairs” of the main text, the occurrence probability of this event cannot be fully captured by observed covariates. In this case, if this event can affect the outcome, its contribution to the outcome will be more (or less) substantial within the treated group after the treatment (i.e., intervention) than that within the control group (Shadish, 2010; West and Thoemmes, 2010). For example, areas experiencing sharp drops in malaria prevalence might also be more likely to experience other events (e.g., sharp drops in the prevalence of other infectious diseases) that can contribute to decreasing the low birth weight rate.”‘

As also mentioned above, we have proposed a sensitivity analysis framework (on pages 14 and 33 of the revised manuscript) with considering a hypothetical unobserved covariate (unmeasured confounding variable or event) U in the outcome model. On page 33 of the revised manuscript, we have discussed how our sensitivity analysis framework covers Perspective 3 which corresponds to your Comment 4:

“For Perspective 3: When setting p1≠0, the hypothetical unobserved covariate *U* in our sensitivity analysis model can also be regarded as some event of which the occurrence probability varies across the treated group and the control group and is not associated with observed covariates. Meanwhile, by setting some p2≠0, the contribution of that event to the low birth weight rate differs across the treated group and the control group as that event occurs more (or less) frequently in the treated group. Therefore, our sensitivity analyses also cover Perspective 3 of the potential hidden bias.”

As also mentioned our sensitivity analyses results and the corresponding interpretations can be found in the “Results of the sensitivity analyses” section of the main text (on page 18 of the revised manuscript) and in the “Detailed results of the sensitivity analyses” section in Appendix 3 (on pages 34 and 35 of the revised manuscript).

References

Angrist, J. D. and Pischke, J.-S. (2008). Mostly Harmless Econometrics: An Empiricist’s Companion. Princeton University Press.

Ashenfelter, O. C. and Card, D. (1984). Using the longitudinal structure of earnings to estimate the effect of training programs.

Ayele, D. G., Zewotir, T. T., and Mwambi, H. G. (2013). The risk factor indicators of malaria in ethiopia. International Journal of Medicine and Medical Sciences, 5(7):335– 347.

Baragatti, M., Fournet, F., Henry, M.-C., Assi, S., Ouedraogo, H., Rogier, C., and Salem, G. (2009). Social and environmental malaria risk factors in urban areas of ouagadougou, burkina faso. Malaria Journal, 8(1):1–14.

Basu, P. and Small, D. S. (2020). Constructing a more closely matched control group in a difference-in-differences analysis: its effect on history interacting with group bias. Observational Studies, 6:103–130.

Bhatt, S., Weiss, D., Cameron, E., Bisanzio, D., Mappin, B., Dalrymple, U., Battle, K., Moyes, C., Henry, A., Eckhoff, P., Wenger, E., Briët, O., Penny, M., Smith, T., Bennett, A., Yukich, J., Eisele, T., Griffin, J., Fergus, C., Lynch, M., Lindgren, F., Cohen, J., Murray, C., Smith, D., Hay, S., Cibulskis, R., and Gething, P. (2015). The effect of malaria control on Plasmodium falciparum in africa between 2000 and 2015. Nature, 526(7572):207.

Card, D. and Krueger, A. B. (2000). Minimum wages and employment: a case study of the fast-food industry in new jersey and pennsylvania: reply. American Economic Review, 90(5):1397–1420.

Carpenter, J. and Kenward, M. (2012). Multiple imputation and its application. John Wiley and Sons.

Chen, X.-K., Wen, S. W., Sun, L.-M., Yang, Q., Walker, M. C., and Krewski, D. (2009). Recent oral contraceptive use and adverse birth outcomes. European Journal of Obstetrics and Gynecology and Reproductive Biology, 144(1):40–43.

Doyle, O., Hegarty, M., and Owens, C. (2018). Population-based system of parenting support to reduce the prevalence of child social, emotional, and behavioural problems: difference-in-differences study. Prevention Science, 19(6):772–781.

Elizabeth Heger Boyle, M. K. and Sobek, M. (2019). Minnesota Population Center and ICF International.

Fakhouri, T. H., Martin, C. B., Chen, T.-C., Akinbami, L. J., Ogden, C. L., PauloseRam, R., Riddles, M. K., Van de Kerckhove, W., Roth, S. B., Clark, J., Mohadjer, L. K., and Fay, R. E. (2020). An investigation of nonresponse bias and survey location variability in the 2017-2018 national health and nutrition examination survey. Vital and Health statistics. Series 2, Data Evaluation and Methods Research, (185):1–36.

Gemperli, A., Vounatsou, P., Kleinschmidt, I., Bagayoko, M., Lengeler, C., and Smith, T. (2004). Spatial patterns of infant mortality in mali: the effect of malaria endemicity. American Journal of Epidemiology, 159(1):64–72.

Grace, K., Davenport, F., Hanson, H., Funk, C., and Shukla, S. (2015). Linking climate change and health outcomes: Examining the relationship between temperature, precipitation and birth weight in africa. Global Environmental Change, 35:125–137.

Hasegawa, R. B., Webster, D. W., and Small, D. S. (2019). Evaluating missouri’s handgun purchaser law: a bracketing method for addressing concerns about history interacting with group. Epidemiology, 30(3):371–379.

Heckman, J. J., Ichimura, H., and Todd, P. E. (1997). Matching as an econometric evaluation estimator: Evidence from evaluating a job training programme. The Review of Economic Studies, 64(4):605–654.

Heckman, J. J. and Robb, R. J. (1985). Alternative methods for evaluating the impact of interventions: An overview. Journal of Econometrics, 30(1-2):239–267.

ICF (2019). 2004-2017. demographic and health surveys (various) [datasets]. funded by usaid. rockville, maryland: Icf [distributor]. Technical report.

Krefis, A. C., Schwarz, N. G., Nkrumah, B., Acquah, S., Loag, W., Sarpong, N., AduSarkodie, Y., Ranft, U., and May, J. (2010). Principal component analysis of socioeconomic factors and their association with malaria in children from the ashanti region, ghana. Malaria Journal, 9(1):1–7.

MAP (2020). Malaria atlas project. The MAP Group.

Meng, X.-L. (1994). Multiple-imputation inferences with uncongenial sources of input. Statistical Science, pages 538–558.

Padhi, B. K., Baker, K. K., Dutta, A., Cumming, O., Freeman, M. C., Satpathy, R., Das, B. S., and Panigrahi, P. (2015). Risk of adverse pregnancy outcomes among women practicing poor sanitation in rural india: a population-based prospective cohort study. PLoS Medicine, 12(7):e1001851.

Roberts, D. and Matthews, G. (2016). Risk factors of malaria in children under the age of five years old in uganda. Malaria Journal, 15(1):1–11.

Rosenbaum, P. R. (2010). Design of Observational Studies, volume 10. Springer.

Rosenbaum, P. R. and Rubin, D. B. (1983). The central role of the propensity score in observational studies for causal effects. Biometrika, 70(1):41–55.

Rosenbaum, P. R. and Silber, J. H. (2009). Amplification of sensitivity analysis in matched observational studies. Journal of the American Statistical Association, 104(488):1398–1405.

Rubin, D. B. (1987). Multiple Imputation for Nonresponse in Surveys. John Wiley and Sons.

Rubin, D. B. (1996). Multiple imputation after 18+ years. Journal of the American statistical Association, 91(434):473–489.

Rubin, D. B. (2007). The design versus the analysis of observational studies for causal effects: parallels with the design of randomized trials. Statistics in Medicine, 26(1):20– 36.

Sahn, D. E. and Stifel, D. C. (2003). Urban–rural inequality in living standards in africa. Journal of African Economies, 12(4):564–597.

Shadish, W. R. (2010). Campbell and rubin: A primer and comparison of their approaches to causal inference in field settings. Psychological Methods, 15(1):3.

St.Clair, T. and Cook, T. D. (2015). Difference-in-differences methods in public finance. National Tax Journal, 68(2):319–338.

Stuart, E. A. (2010). Matching methods for causal inference: A review and a look forward. Statistical Science, 25(1):1.

Sulyok, M., Rückle, T., Roth, A., Mürbeth, R. E., Chalon, S., Kerr, N., Samec, S. S., Gobeau, N., Calle, C. L., Ibáñez, J., Sulyok, Z., Held, J., Gebru, T., Granados, P., Brückner, S., Nguetse, C., Mengue, J., Lalremruata, A., Sim, B. K. L., Hoffman, S. L., Möhrle, J. J., Kremsner, P. G., and Mordmüller, B. (2017). DSM265 for Plasmodium falciparum chemoprophylaxis: a randomised, double blinded, phase 1 trial with controlled human malaria infection. The Lancet Infectious Diseases, 17(6):636–644.

West, S. G. and Thoemmes, F. (2010). Campbell’s and rubin’s perspectives on causal inference. Psychological Methods, 15(1):18.

WHO (2008). Global malaria action plan 1 (2000–2015). Roll Back Malaria Partnership/World Health Organization.